

# Inverted distribution of ductile deformation in the relatively "dry" middle crust across the Woodroffe Thrust, central Australia

Sebastian Wex[1], Neil S. Mancktelow[1], Friedrich Hawemann[1], Alfredo Camacho[2], Giorgio Pennacchioni[3]

[1]Department of Earth Sciences, ETH Zurich, Sonneggstrasse 5, 8092 Zurich, Switzerland
[2]Department of Geological Sciences, University of Manitoba, 125 Dysart Rd, Winnipeg, Manitoba, R3T 2N2, Canada
[3]Department of Geosciences, University of Padova, Via Gradenigo 6, 35131 Padova, Italy

*Correspondence to*: Neil S. Mancktelow (neil.mancktelow@erdw.ethz.ch)

**Abstract.** Thrust fault systems typically distribute shear strain preferentially into the hanging wall rather than the footwall. In this paper, we present a regional-scale example that does not fit this model. The Woodroffe Thrust developed due to
intracontinental shortening during the Petermann Orogeny (ca. 560-520 Ma) in central Australia. It is interpreted to be at least 600 km long in its general E-W strike direction, with an approximate top-to-north minimum relative displacement of 60-100 km. The associated mylonite zone is most broadly developed in the footwall. The immediate hanging wall was only marginally involved in mylonitization, as can be demonstrated from the contrasting thorium signatures of the upper amphibolite facies footwall and the granulite facies hanging wall protoliths. Thermal weakening cannot account for such an inverse deformation
gradient, as syn-deformational P-T estimates for the Petermann Orogeny in the hanging wall and footwall from the same locality are very similar. The distribution of pseudotachylytes, which act as preferred nucleation sites for shear deformation, also cannot provide an explanation, since these are prevalent in the immediate hanging wall. The most likely reason for the inverted deformation gradient across the Woodroffe Thrust is water-assisted weakening due to the increased, but still limited, presence of aqueous fluids in the footwall. On the contrary, the presence or absence of aqueous fluids does not appear to be
linked to the regional variation in mylonite thickness, which generally increases with increasing metamorphic grade.

## 1 Introduction

Continental fault systems with displacements on the order of several tens to hundreds of kilometres generally show an asymmetric mylonite distribution across the main fault horizon that is opposite for reverse faults or thrusts and normal faults or detachments. Fault zones are predicted to become more viscous and broaden with depth (e.g., Fossen and Cavalcante, 2017;
Handy et al., 2007; Mancktelow, 1985; Passchier and Trouw, 2005; Platt and Behr, 2011b). Juxtaposition of initially different crustal levels should therefore result in a geometry that, for a thrust, preferentially preserves the broader ductile mylonite zone in the hanging wall, whereas, for a detachment, it should be in the footwall (e.g., Mancktelow, 1985, his Fig. 11). This model is valid for many large-scale fault systems, for example: the Simplon Fault, central European Alps (Mancktelow, 1985), the Alpine Fault, New Zealand (Cooper and Norris, 1994; Sibson et al., 1981), the Grizzly Creek shear zone, Colorado (Allen and
Shaw, 2011) and the Whipple mountains detachment fault, southwestern U.S. (Davis, 1988; Davis and Lister, 1988). The

midcrustal Woodroffe Thrust of central Australia (Major, 1970) is an example that does not fit this model and predominantly developed a broader mylonite zone in the footwall (Bell and Etheridge, 1976; Camacho et al., 1995; Flottmann et al., 2004). An interpretation of the Woodroffe Thrust as an original detachment that was later re-oriented and exploited as a thrust can be excluded, both because the metamorphic grade decreases in the direction of tectonic transport and because field mapping

shows that the fault zone steepens and ramps down towards the internal part of the orogen, against the transport direction (Wex et al., 2017). Passive transport and thermal weakening also cannot account for the inverse deformation gradient, as there is no evidence for late brittle movement on the thrust plane and syn-deformational P-T estimates in the hanging wall and footwall from the same locality are very similar (Wex et al., 2017). Bell and Etheridge (1976) and Camacho et al. (1995) proposed that the inverted distribution of ductile deformation is explained by the difference in bulk water content between the upper

amphibolite facies (1.0 wt%) footwall and the granulite facies (0.2 wt%) hanging wall, reflecting the metamorphic conditions in the protolith prior to thrusting. Similarly, the preferential formation of shear zones in regions where the host rock mineralogy had previously been modified by fluid-rock interaction has, for example, also been documented in the Whipple mountains detachment fault in SE California (Selverstone et al., 2012) and the Neves area of the Tauern Window in the eastern Alps (Mancktelow and Pennacchioni, 2005). In this paper, we quantify the control of host rock lithology and potential fluid activity

on the distribution of ductile deformation across the Woodroffe Thrust, in an attempt to critically test the local findings of Bell and Etheridge (1976) from the Amata area (western edge of Fig. 1) on a more regional scale (Fig. 1). We also investigate the effect of varying metamorphic temperatures and fluid conditions on the variation in mylonite thickness over a distance of ca. 60 km parallel to the direction of thrusting.

## 2 Geology

The crustal-scale Woodroffe Thrust in the Musgrave Block of central Australia (Major, 1970) is developed over an approximate E-W strike length generally interpreted to exceed 600 km. It separates the Mulga Park Subdomain in the footwall from the Fregon Subdomain in the hanging wall (Edgoose et al., 1993; Major and Conor, 1993). Exposure of the Woodroffe Thrust is poor to inexistent in the proposed western (e.g., Stewart, 1995, 1997) and eastern (e.g., Edgoose et al., 2004) prolongations, but is generally excellent for ca. 150 km in the central Musgrave Block (Bell, 1978; Camacho et al., 1995;

Collerson et al., 1972; Wex et al., 2017), where the current study was conducted. In this region, both footwall and hanging wall predominantly consist of granitoids (more common in the footwall) and quartzo-feldspathic gneisses (more common in the hanging wall), with subordinate metadolerites, mafic gneisses and metapelites (Fig. 1). Rare quartzites, amphibolites and schists are restricted to the footwall (Camacho and Fanning, 1995; Collerson et al., 1972; Major, 1973; Major and Conor, 1993; Scrimgeour and Close, 1999; Young et al., 2002). Protoliths are inferred to have been felsic volcanics, sediments and intrusives

with depositional or emplacement ages around ca. 1550 Ma (Camacho, 1997; Camacho and Fanning, 1995; Gray, 1977, 1978; Gray and Compston, 1978; Maboko et al., 1991; Major and Conor, 1993; Sun and Sheraton, 1992). These rocks were regionally







**Figure 1: Geological map of the central Musgrave Block (modified after Major et al., 1967; Sprigg et al., 1959; Young et al., 2002).**



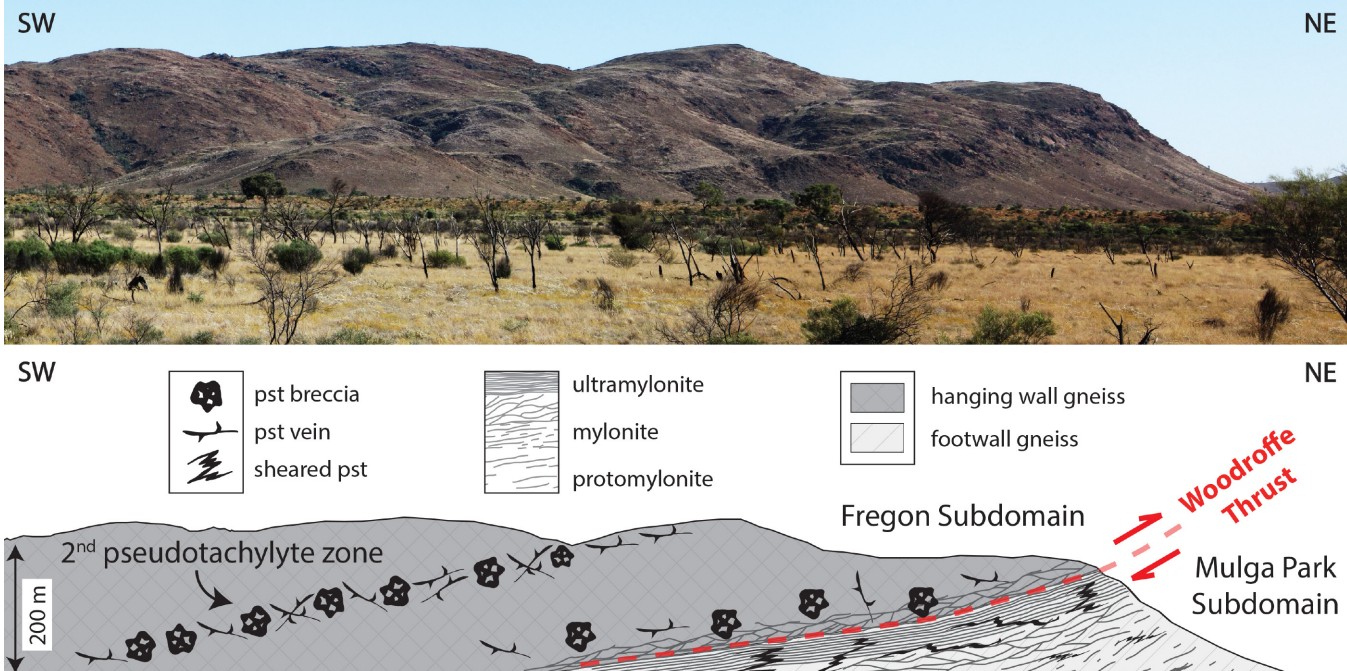

**Figure 2: Photograph and schematic sketch of a cross section through the Woodroffe Thrust at Kelly Hills. Ductile deformation**
**almost entirely localized in the immediate footwall (Mulga Park Subdomain), developing a sequence of protomylonites, mylonites**
**and ultramylonites, with the degree of mylonitization decreasing into the footwall. In contrast, the mostly unaffected or only weakly**
**foliated hanging wall (Fregon Subdomain) is characterized by ubiquitous and voluminous pseudotachylyte veins and breccias.**
**Further into the hanging wall, a dip slope, characterized by a second zone of highly abundant unsheared pseudotachylyte, has also**
**been documented. Photograph coordinates: 131.45077, -25.89823.**

metamorphosed into upper amphibolite facies (Mulga Park Subdomain) to granulite facies (Fregon Subdomain) gneisses
during the ca. 1200 Ma Musgravian Orogeny (Camacho, 1997; Camacho and Fanning, 1995; Gray, 1978; Maboko et al., 1991;
Sun and Sheraton, 1992) and syn- to posttectonically intruded by the Pitjantjatjara Supersuite granitoids between ca. 1170-
1130 Ma (Camacho, 1997; Camacho and Fanning, 1995; Scrimgeour et al., 1999; Smithies et al., 2011). Subsequently, the
area experienced bimodal magmatism (Giles Complex, including the Alcurra Dolerite swarm) between ca. 1080-1050 Ma and
mafic magmatism (Amata Dolerite) at ca. 800 Ma (Ballhaus and Glikson, 1995; Camacho et al., 1991, 1997; Clarke et al.,
1995; Edgoose et al., 1993; Sun et al., 1996; Zhao et al., 1994; Zhao and McCulloch, 1993). In the area considered in the
current study, large Giles Complex gabbro-norite intrusions are restricted to the Fregon Subdomain. In parts of the field area
(locations 11-13 in Fig. 1), mylonites with top-to-west kinematics, postdating the emplacement of dolerite dykes but preceding
the main Woodroffe Thrust mylonites, have been recognized but remain largely undocumented (Wex et al., 2017). Aside from
this local deformation phase, the central Musgrave Block remained virtually unaffected by tectonic events between the
Musgravian Orogeny at ca. 1200 Ma and the Petermann Orogeny at ca. 560-520 Ma



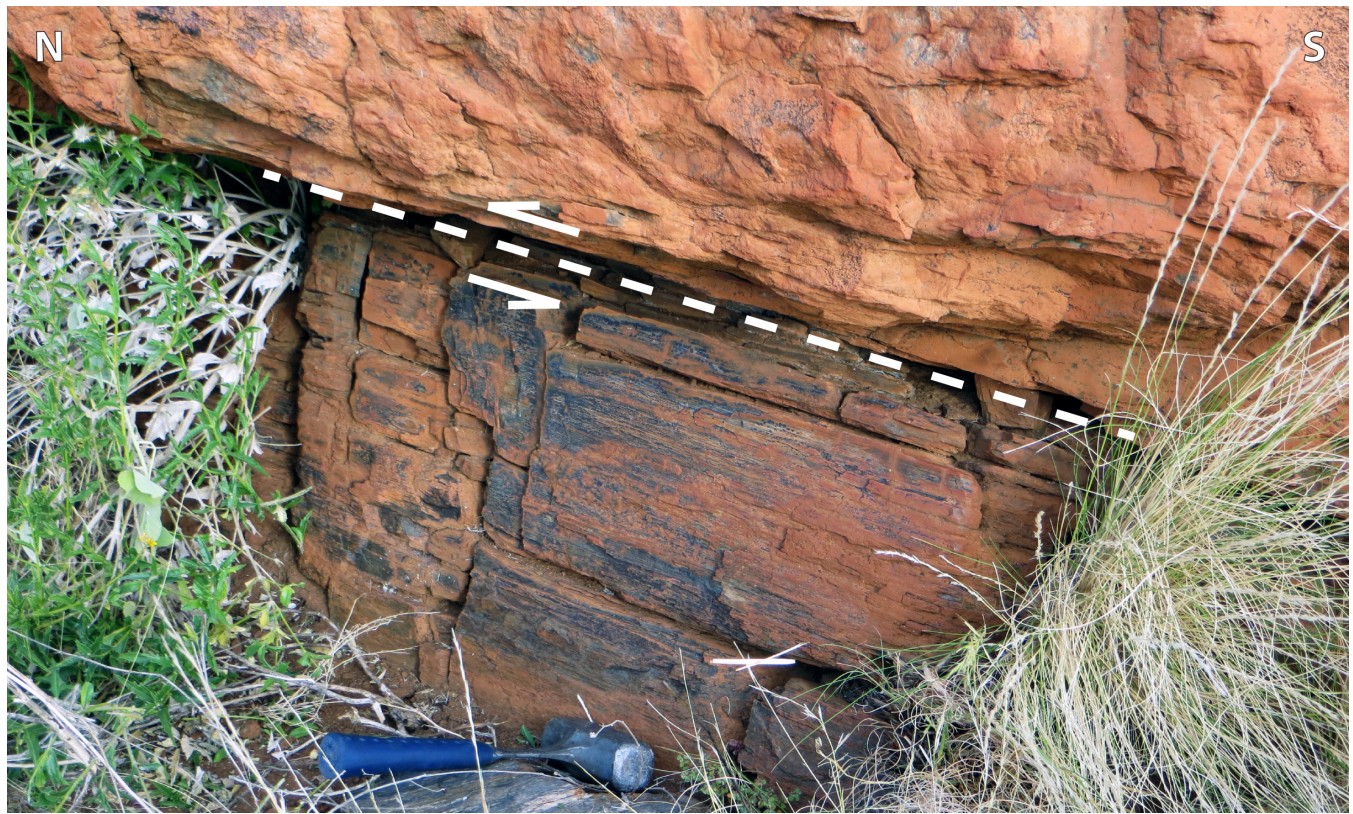

**Figure 3: Sharp contact between the ultramylonites of the Woodroffe Thrust (below dashed line) and the largely undeformed felsic granulite in the hanging wall (above dashed line). Photograph looking perpendicular to the direction of thrusting. Outcrop SW13-135 (coordinates: 131.87939, -26.21188; location 12 in Fig. 1).**

(Camacho and Fanning, 1995; Maboko et al., 1992). The Petermann Orogeny produced a number of large-scale mylonitic shear zones, amongst which the Woodroffe Thrust is the most prominent. Ductile deformation during top-to-north thrusting along the Woodroffe Thrust was largely accommodated in the Mulga Park Subdomain (Bell and Etheridge, 1976; Camacho et al., 1995; Flottmann et al., 2004) and is characterized by mylonites with varying degrees of strain, ranging from protomylonites to ultramylonites (Fig. 2), anastomosing around low-strain domains on the metre- to kilometre-scale. These mylonites preserve an annealed and a non-annealed microstructure, interpreted to record lower crustal (ca. 650 °C and 1.0-1.3 GPa) and midcrustal (ranging from ca. 520-620 °C and 0.8-1.1 GPa) shearing stages of the Woodroffe Thrust (Wex et al., 2017). Stable mineral assemblages in felsic units comprise (decreasing modal abundance from left to right) Qz + Pl + Kfs + Grt + Bt + Ilm ± Ep ± Ms ± Ky ± Cpx ± Hbl ± Rt ± Ttn ± Mag ± Cal, whereas mafic units consist of Pl + Cpx + Grt + Ilm ± Rt ± Opx ± Bt ± Hbl ± Ky ± Mag ± Qz ± Kfs ± Cal (Wex et al., 2017). Mineral abbreviations are after Whitney and Evans (2010). The degree of mylonitization progressively decreases into the footwall but shows a very abrupt transition into the immediate, dominantly brecciated hanging wall (Figs. 2, 3), which is characterized by ubiquitous and voluminous pseudotachylyte veins and breccias (Camacho et al., 1995; Lin et al., 2005). Even though this upper boundary of the mylonites is discrete or rapidly transitional



in the field (Fig. 3), it does not necessarily represent the original boundary between the Mulga Park (footwall) and Fregon
(hanging wall) Subdomains. In the Amata area (western edge of Fig. 1), Bell (1978) reported up to 250 m of marginal hanging
wall reworking into the mylonite zone. However, it remains uncertain how this value was exactly determined, since hanging
wall and footwall mylonites are very similar in their field appearance.

## 3 Methods and general approach

The hanging wall, footwall and numerous transects across the Woodroffe Thrust have been studied and sampled at the locations
reported in Fig. 1. Thin sections of the sampled mylonites were cut perpendicular to the foliation and parallel to the stretching
lineation and analysed using standard polarized light and scanning electron microscopy. Firstly, the distribution of ductile
deformation along and across the Woodroffe Thrust was characterized by quantifying (1) the regional variation in the
maximum thickness of the mylonitic zone and (2) the associated degree of hanging wall and footwall reworking. Secondly,
field and thin section observations were compiled to assess in a qualitative manner (3) the presence/absence of aqueous fluids
during deformation, and (4) the regional variability in modal abundance of hydrous minerals in felsic units. Potential
correlations between parameters (1) to (4) are then discussed. Sample/outcrop coordinates are given in the world geodetic
system (WGS) 1984. Orientation measurements of structural elements are corrected for magnetic declination. A detailed
description of all utilized methods is given in the Supporting Information S1.

## 4 Mylonite thickness

Field observations indicate that the thickness of the Woodroffe Thrust mylonites is variable. In a section perpendicular to
strike, the thickness ($T$) of the mylonitic zone across the Woodroffe Thrust was calculated by trigonometry (Fig. 4) from: (i)
the angle of dip ($\alpha$) of the thrust (measured in the field and averaged for each transect); (ii) the respective difference in elevation
($p$) (derived from the 30x30 m digital elevation model ASTER) between the lower and upper structural boundaries of the
mylonites to the unsheared country rocks (determined from field observations and remote sensing); and (iii) the apparent
thickness ($q$) (derived from remote sensing). The upper boundary of the mylonitic zone is easily recognized (Fig. 3), whereas
the lower boundary is generally less well defined due to its gradual and irregular nature (Fig. 2), with high-strain shear zones
surrounding less to little deformed low-strain domains on the metre- to kilometre-scale. Errors for parameters $\alpha$ and $q$ are
considered negligible whereas the 30x30 m resolution of the digital elevation model is prone to introduce an uncertainty on
the order of 10-20 m. The geometrical arrangement for the estimate of $T$ was the same for locations 2-8 and 12-14 (Fig. 4a),
but slightly different for location 11 (Fig. 1), where the lower structural boundary of the mylonites was at a higher elevation
than the upper boundary (Fig. 4b). The thickness of the mylonitic zone at location 1 (Fig. 1) was not calculated, because the
Woodroffe Thrust is only exposed along strike. The results for all other studied transects are summarized in Table 1 and,



independent of local variability, indicate a gradual increase in mylonite thickness from the northern locations 2-8 (average 120 m) via the central locations 11-12 (ca. 300 m) to the southern locations 13-14 (ca. 600 m).


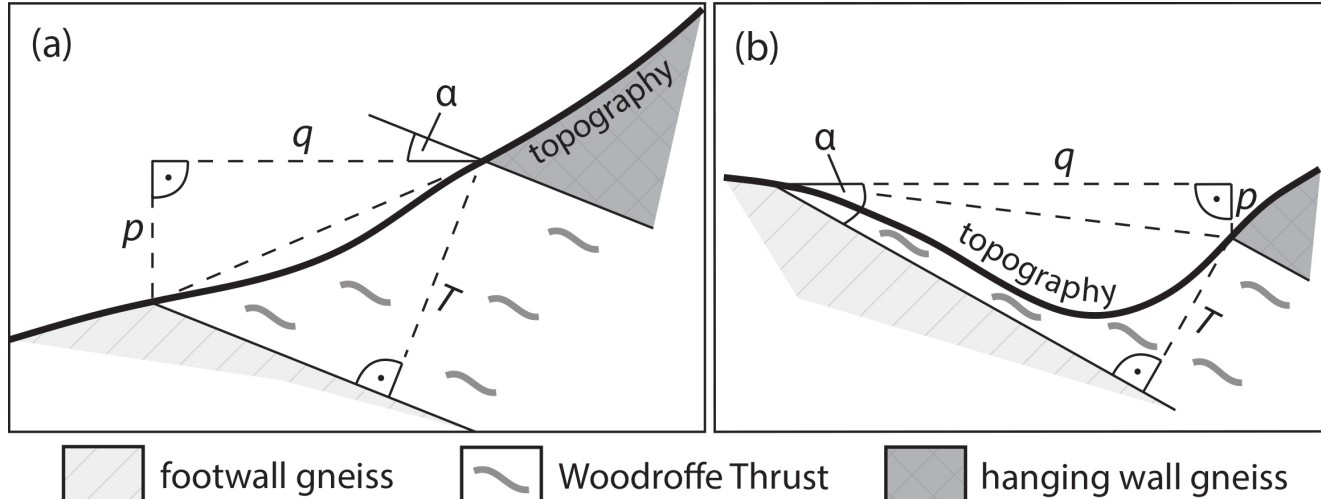

**Figure 4: Schematic illustration of the trigonometry applied to quantify the true thickness of the Woodroffe Thrust mylonitic zone in a section perpendicular to strike. The parameters are defined in the main text. (a) Geometry applicable to locations 2-8 and 12-14 of Fig. 1. (b) Geometry applicable to location 11 of Fig. 1.**


| location | coordinates (WGS 1984) | | trigonometrical parameters | | | |
|---|---|---|---|---|---|---|
| (Fig. 1) | longitude | Latitude | $\alpha$ (°) | $p$ (m) | $q$ (m) | $T$ (m) |
| 2 | 131.454 | -25.845 | 27 | 22 | 150 | 88 |
| 3 | 131.452 | -25.855 | 43 | 54 | 190 | 169 |
| 4 | 131.449 | -25.873 | 36 | 23 | 160 | 113 |
| 5 | 131.442 | -25.904 | 21 | 44 | 130 | 88 |
| 6 | 132.143 | -25.992 | 29 | 62 | 160 | 132 |
| 7 | 131.644 | -25.963 | 17 | 110 | 250 | 178 |
| 8 | 131.663 | -25.999 | 10 | 60 | 130 | 82 |
| 11 | 131.926 | -26.177 | 28 | 61 | 800 | 322 |
| 12 | 131.879 | -26.212 | 24 | 153 | 430 | 315 |
| 13 | 131.844 | -26.253 | 37 | 124 | 850 | 611 |
| 14 | 131.774 | -26.308 | 25 | 128 | 1200 | 623 |

**Table 1: Angle of dip ($\alpha$), elevation difference ($p$), apparent thickness ($q$) and true thickness ($T$) of the Woodroffe Thrust mylonites in the central Musgrave Block.**



**Figure 5: Airborne thorium (ppm) map of the central Musgrave Block. Data derived from the Australian Geophysical Archive Data Delivery System (GADDS) under www.ga.gov.au/gadds (Percival, 2010). The hanging wall Fregon Subdomain is generally depleted compared to the footwall Mulga Park Subdomain. Granitoid intrusives of the Pitjantjatjara Supersuite often have higher-thorium content, particularly in the footwall (one example surrounded by dashed lines), but also in the southern part of the hanging wall. Lower-thorium signatures forming diffuse tongues extending northward into the footwall are attributed to alluvial wash derived from the hanging wall. The distinction in thorium content is no longer evident for the northernmost klippe of the hanging wall in the Kelly Hills area (locations 1-5) and the northernmost part of the Mt. Fraser klippe (location 7).**

## 5 Degree of hanging wall and footwall mylonitization

Mylonites derived from the hanging wall and footwall of the Woodroffe Thrust are very similar in both their field appearance and microstructural-petrographical characteristics. The high degree of recrystallization and general lack of porphyroclasts in the uppermost mylonites at locations 4, 6-8 and 12-14 (Fig. 1) suggests that these samples were potentially derived from the pseudotachylyte-rich Fregon Subdomain, hence indicating local hanging wall reworking. To establish whether mylonites were derived from either the Mulga Park or the Fregon Subdomains, we utilized their thorium concentrations, based on the general



observation that hanging wall rocks are depleted in thorium compared to footwall rocks (Fig. 5). This contrast is due to (1)
dehydration and melting reactions during the earlier ca. 1200 Ma Musgravian Orogeny, which depleted the granulite facies
hanging wall to a greater degree than the upper amphibolite facies footwall (Heier and Adams, 1965; Lambert and Heier, 1967,
1968), (2) the predominance of granitoids rather than gneisses in the footwall, and (3) the fact that low-thorium Giles Complex
gabbro-noritic intrusions are often exposed in the immediate hanging wall of the Woodroffe Thrust. Deformation and
metamorphism during mylonitization did not significantly alter the original thorium content of the rocks, since the thorium-
bearing phases, such as zircon, allanite, monazite and apatite did not break down during the Petermann Orogeny. Consequently,
the original variation in thorium between the granulite and amphibolite facies rocks was preserved (Fig. 5). Anomalies in this
broad pattern are present locally and can be attributed to hanging wall-derived alluvial sediments in the immediate footwall
(lower-thorium anomaly) and to granitoid intrusions of the Pitjantjatjara Supersuite and Giles Complex (higher-thorium
anomaly), which are syn- to post-Musgravian upper amphibolite to granulite facies metamorphism (Scrimgeour and Close,
1999; Young et al., 2002) (Fig. 5). The contrast between a lower-thorium hanging wall and a higher-thorium footwall is well-
defined on the airborne thorium map for the central and southern locations 6, 8 and 11-14, but less evident in the northern
locations 1-5 and 7 (Fig. 5). Thorium measurements were carried out on thin section chips of felsic gneisses and granitoids via
γ-ray spectrometry. The method is outlined in detail in the Supporting Information S1.

**5.1 Thorium concentration in felsic units**

Thorium contents of felsic gneisses and granitoids (Table 2) have been compiled from the current study as well as from
previous studies of nearby areas (Camacho, 1997; Young et al., 2002). These data have been grouped into (1) the central and
southern locations (6, 8, and 11-14) and (2) the northern locations (1-5, and 7), based on the airborne thorium map, as
introduced above.

In group (1), samples unequivocally attributed to either the hanging wall or footwall of the Woodroffe Thrust were grouped
together. This was done on the basis of their geographical position, with respective samples originating either from far into the
non-mylonitized hanging wall and footwall or from the lowermost protomylonitic part of the Woodroffe Thrust. In the hanging
wall, thorium concentrations range from 1 to 28 ppm in felsic gneisses and up to 63 ppm in granitic intrusions (higher-thorium
anomalies), but are in both cases usually lower than 8 ppm. In the footwall, concentrations vary between 2 and 195 ppm and
are typically higher than 10 ppm (Table 2). These concentrations are in accord with the regional-scale contrast in thorium
concentrations across the Woodroffe Thrust in the central and southern locations 6, 8 and 11-14 (Fig. 5), as well as with the
results of Lambert and Heier (1968), who determined concentrations of 2.1 ppm for the granulite facies hanging wall and 11
ppm for the upper amphibolite facies footwall. Based on the compilation in Table 2, samples with thorium content <8 ppm
were assigned to the hanging wall and samples with higher values to the footwall of the Woodroffe Thrust. Five different
transects were investigated (locations 8 and 11-14 in Fig. 5), generally comprising samples taken close to the boundary between
the mylonites and unsheared rocks of the Fregon Subdomain (typically the uppermost few tens to one hundred metres). For



the majority of samples, the assignment was straightforward since the inferred hanging wall samples are extremely low in thorium (<3 ppm), whereas most inferred footwall samples have values >8 ppm. Exceptions are samples SW14-243 and SW13-159, which have intermediate concentrations of 6 ppm and 8 ppm, respectively. Both samples were assigned to the footwall since subsequent samples further towards the hanging wall, respectively SW14-244 and SW13-161, could clearly still be attributed to the footwall. Alternatively, samples SW14-243 and SW13-159 could reflect imbrication of the Mulga Park and

Fregon Subdomains, but this is not supported by any field observation.

In group (2), four different transects were investigated (locations 2-4 and 6 in Fig. 5). Thorium concentrations vary between 11 and 49 ppm (Table 2) but do not allow a clear distinction between samples derived from the footwall and hanging wall in a manner similar to the central and southern locations. This result is in accord with the airborne thorium concentrations, which also do not indicate a significant jump across the Woodroffe Thrust in these more northerly locations (Fig. 5).




### central and southern studied locations

**hanging wall of the Woodroffe Thrust**

| location (Fig. 1) | sample | lithology | longitude | latitude | data source | Th (ppm) | |
|---|---|---|---|---|---|---|---|
| 9 | SW14-029A | granite | 131,74496 | -26,00093 | 1 | 6 | |
| 9 | SW14-025 | granite | 131,73269 | -26,01569 | 1 | 3 | |
| 9 | SW14-030B | granite | 131,74295 | -26,00222 | 1 | 5 | |
| 17 | FW13-173 | granite | 131,54138 | -26,34104 | 1 | 3 | |
| 18 | FW13-228 | granite | 131,56005 | -26,37413 | 1 | 28 | |
| - | MP-2 | granite | 131,74845 | -25,99871 | 3 | 10 | |
| - | MP94/500 | granite | 131,73549 | -25,99535 | 3 | 4 | |
| - | MP94/501 | granite | 131,74105 | -25,99389 | 3 | 11 | |
| - | W-12 | granite | 131,55670 | -26,38583 | 2 | 19 | |
| - | W-34d | granite | 131,52310 | -26,34528 | 2 | 6 | |
| - | W-70 | granite | 131,63910 | -26,40186 | 2 | 4 | |
| - | W-96 | granite | 131,69440 | -26,38511 | 2 | 8 | |
| - | W-104 | granite | 131,70010 | -26,38119 | 2 | 4 | |
| - | W-127 | granite | 131,90190 | -26,43750 | 2 | 7 | correlation based on geographical location |
| - | W-146 | granite | 131,87670 | -26,35469 | 2 | 6 | |
| - | W-148 | granite | 131,88030 | -26,34653 | 2 | 10 | |
| - | W-199a | granite | 131,77080 | -26,47550 | 2 | 54 | |
| - | W-199b | granite | 131,77080 | -26,47550 | 2 | 63 | |
| - | E-37 | charnockite | 132,27078 | -26,19010 | 2 | 6 | |
| - | E-38 | charnockite | 132,26899 | -26,18286 | 2 | 3 | |
| - | E-39 | charnockite | 132,14968 | -26,22028 | 2 | 2 | |
| - | W-32 | charnockite | 131,46100 | -26,29419 | 2 | 26 | |
| - | W-129 | charnockite | 131,89100 | -26,38531 | 2 | 8 | |
| 16 | NW13-043 | felsic gneiss | 131,49727 | -26,28058 | 1 | 4 | |
| 16 | SW13-032 | felsic gneiss | 131,49771 | -26,28135 | 1 | 2 | |
| 19 | NW13-016B | felsic gneiss | 131,71053 | -26,38530 | 1 | 28 | |
| 20 | NW13-026 | felsic gneiss | 131,66087 | -26,41038 | 1 | 20 | |
| 20 | NW13-030 | felsic gneiss | 131,66258 | -26,40675 | 1 | 1 | |
| - | E-27 | felsic gneiss | 132,14091 | -26,18857 | 2 | 3 | |
| - | E-47 | felsic gneiss | 132,04583 | -26,29111 | 2 | 2 | |
| - | MP97/513 | felsic gneiss | 131,66461 | -25,97300 | 3 | 12 | |
| - | W-16dg | felsic gneiss | 131,52810 | -26,26806 | 2 | 6 | |
| - | W-16ug | felsic gneiss | 131,52810 | -26,26806 | 2 | 8 | |
| - | W-18 | felsic gneiss | 131,53810 | -26,27472 | 2 | 21 | |
| - | W-122adg | felsic gneiss | 131,69830 | -26,28992 | 2 | 3 | |
| - | W-122aug | felsic gneiss | 131,69830 | -26,28992 | 2 | 8 | |
| - | W-140b | felsic gneiss | 131,84610 | -26,43422 | 2 | 5 | |
| - | W-142a | felsic gneiss | 131,84050 | -26,44422 | 2 | N/D | |
| 8 | SW13-191 | felsic gneiss | 131,66264 | -25,99911 | 1 | N/D < 2 | |
| 11 | SW14-181A | felsic gneiss | 131,92595 | -26,17664 | 1 | N/D < 2 | |
| 11 | SW14-182 | felsic gneiss | 131,92564 | -26,17681 | 1 | 1 | |
| 12 | NW13-187B | felsic gneiss | 131,87917 | -26,21151 | 1 | 1 | corr. based on Th conc. |
| 12 | SW13-135A | felsic gneiss | 131,87939 | -26,21188 | 1 | 2 | |
| 12 | SW13-135B | felsic gneiss | 131,87939 | -26,21188 | 1 | 2 | |
| 12 | NW13-189 | felsic gneiss | 131,87935 | -26,21191 | 1 | 2 | |
| 13 | SW14-246A | felsic gneiss | 131,84444 | -26,25262 | 1 | 2 | |
| 14 | SW13-164 | felsic gneiss | 131,77463 | -26,30772 | 1 | 3 | |

**footwall of the Woodroffe Thrust**

| location (Fig. 1) | sample | lithology | longitude | latitude | data source | Th (ppm) | |
|---|---|---|---|---|---|---|---|
| 12 | SW13-122A | granite | 131,87206 | -26,20364 | 1 | 73 | |
| 12 | SW13-123 | granite | 131,87229 | -26,20379 | 1 | 7 | |
| 12 | NW13-184 | granite | 131,87823 | -26,20914 | 1 | 16 | |
| 12 | NW13-185B | granite | 131,87830 | -26,20946 | 1 | 14 | |
| 14 | SW13-151 | granite | 131,77438 | -26,30117 | 1 | 20 | |
| 14 | SW13-153 | granite | 131,77411 | -26,30173 | 1 | N/D < 2 | correlation based on geographical location |
| - | MP94/502 | granite | 131,78418 | -25,95562 | 3 | 55 | |
| - | MP94/503 | granite | 131,62926 | -25,86096 | 3 | 30 | |
| - | BJ96/178 | granite | 131,48641 | -25,78867 | 3* | 18 | |
| - | BJ96/201 | granite | 131,39771 | -25,79945 | 3* | 195 | |
| - | BJ96/276 | granite | 131,44734 | -25,97263 | 3* | 18 | |
| - | MP96/505 | granite | 131,64751 | -25,95006 | 3* | 35 | |
| - | MP96/509 | granite | 131,64996 | -25,94452 | 3 | 12 | |
| 12 | SW13-125 | felsic gneiss | 131,87338 | -26,20470 | 1 | 27 | |
| - | MP97/43 | felsic gneiss | 131,81710 | -25,98836 | 3 | 10 | |
| 8 | GW13-415 | felsic gneiss | 131,66256 | -25,99928 | 1 | 15 | |
| 8 | SW13-193 | felsic gneiss | 131,66256 | -25,99928 | 1 | 16 | |
| 8 | SW13-192 | felsic gneiss | 131,66258 | -25,99915 | 1 | 23 | |
| 11 | SW14-179 | felsic gneiss | 131,92603 | -26,17661 | 1 | 15 | corr. based on Th conc. |
| 12 | SW13-134 | felsic gneiss | 131,87913 | -26,21151 | 1 | 9 | |
| 13 | SW14-243 | felsic gneiss | 131,83942 | -26,25476 | 1 | 8 | |
| 13 | SW14-244 | felsic gneiss | 131,83949 | -26,25472 | 1 | 13 | |
| 14 | SW13-159 | felsic gneiss | 131,77375 | -26,30666 | 1 | 6 | |
| 14 | SW13-161 | felsic gneiss | 131,77445 | -26,30759 | 1 | 14 | |

data source:
| | |
|---|---|
| 1 | this study |
| 2 | Camacho (1997) |
| 3 | Young et al. (2002) |
| * | values erroneously published in Young et al. (2002), but correctly presented here (personal communication A. Camacho) |
| N/D | below detection limit |

**northern studied locations**

| location (Fig. 1) | sample | lithology | longitude | latitude | data source | Th (ppm) |
|---|---|---|---|---|---|---|
| 6 | SW13-321 | granite | 132,14333 | -25,99178 | 1 | 23 |
| 6 | SW13-322 | granite | 132,14340 | -25,99191 | 1 | 49 |
| 3 | NW13-134A | felsic gneiss | 131,45284 | -25,85477 | 1 | 34 |
| 2 | SW13-173A | felsic gneiss | 131,45334 | -25,84543 | 1 | 17 |
| 2 | SW13-173C | felsic gneiss | 131,45334 | -25,84543 | 1 | 31 |
| 3 | FW13-093 | felsic gneiss | 131,45213 | -25,85455 | 1 | 11 |
| 4 | SW14-006 | felsic gneiss | 131,45047 | -25,87174 | 1 | 20 |
| 6 | SW13-323A | felsic gneiss | 132,14344 | -25,99211 | 1 | 19 |
| 6 | SW14-034A | felsic gneiss | 132,14337 | -25,99224 | 1 | 12 |


**Table 2: Thorium concentrations in felsic gneisses and granitoids from the hanging wall and footwall of the Woodroffe Thrust, central Musgrave Block.**



**Figure 6: Sample-specific thorium (Th) concentrations (measured by γ-ray spectrometry) plotted across the Woodroffe Thrust (WT) mylonites at locations 8 and 11-14 (Fig. 1). The original boundary between the lower-thorium hanging wall and higher-thorium footwall is inferred (red line).**

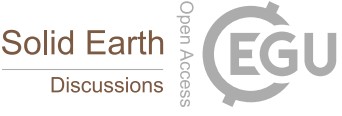

## 5.2 Determination of the hanging wall-footwall boundary

Measured thorium concentrations indicate that in locations 8 and 11-14 (Fig. 5) the uppermost mylonites of the Woodroffe

Thrust developed in the lower-thorium Fregon Subdomain (Table 2). These results are in agreement with the proposed identification of hanging wall reworking based on field appearance. Similar field relationships, such as progressive downwards mylonitization of hanging wall pseudotachylyte breccia, also indicates limited reworking of the Fregon Subdomain at locations 4 and 6 (Fig. 5). However, the lack of a clear contrast in thorium concentrations between footwall and hanging wall in the northern locations precludes any verification of these field observations, as well as any independent determination of the

thickness of hanging wall-derived mylonites in the thrust zone. We therefore excluded these northern transects when using thorium concentration as a proxy to map the original boundary between the hanging wall and footwall (Fig. 6). It was also not possible to precisely determine the boundary at location 13 (Fig. 1), due to a large lateral gap between the last sample assigned to the footwall and the first sample assigned to the hanging wall. In an attempt to quantitatively calculate the degree of hanging wall/footwall reworking into the total mylonite zone, we applied trigonometry based on the geometrical arrangement sketched

in Fig. 4a. The only modification was that the lower mylonite boundary was now defined by the newly reconstructed boundary between hanging wall and footwall. Results are summarized in Table 3. In contrast to the 250 m proposed by Bell (1978) for the Amata area (western edge of Fig. 1), our results indicate that only the lowermost 3 m of the Fregon Subdomain were reworked into the Woodroffe Thrust mylonites at location 8, increasing up to 18-40 m at locations 11-14 (Fig. 1). These values represent 3-6 % of the entire thickness of the Woodroffe Thrust mylonites (Table 1) at locations 8, 11 and 14 and up to 13 %

at location 12 (Fig. 1). However, the difference in elevation ($p$) is not well defined over short distances ($q$) given the limited (30x30 m) resolution of the digital elevation model (ASTER). This can introduce a significant uncertainty into the calculation of the degree of hanging wall reworking. Nevertheless, our analysis clearly shows that the majority of mylonites developed in the Mulga Park Subdomain (footwall) rather than the Fregon Subdomain (hanging wall).

| location | coordinates (WGS 1984) | | trigonometrical parameters | | | |
|---|---|---|---|---|---|---|
| (Fig. 1) | longitude | latitude | $\alpha$ (°) | $p$ (m) | $q$ (m) | $T$ (m) |
| 8 | 132.143 | -25.992 | 10 | 1 | 10 | 3 |
| 11 | 131.926 | -26.177 | 28 | 5 | 30 | 18 |
| 12 | 131.879 | -26.212 | 21 | 22 | 55 | 40 |
| 13 | 131.844 | -26.253 | - | - | - | >0 |
| 14 | 131.774 | -26.308 | 22 | 9 | 30 | 20 |


**Table 3: Angle of dip ($\alpha$), elevation difference ($p$), apparent thickness ($q$) and true thickness ($T$) of the Woodroffe Thrust mylonites derived from the hanging wall.**



## 6 Presence or absence of fluids during mylonitization

The syn-deformational presence or absence of fluids in the study area is established from a series of field and thin section
observations. These include the regional variation in: (1) syntectonic quartz veins, (2) introduction of carbon and (3) plagioclase stability and breakdown.

### 6.1 Quartz veins

Syntectonic quartz veins (Fig. 7a) and associated quartz-rich pegmatite dykes are uncommon throughout the field area, being generally absent in the southern locations 10-15 and only locally present in the northern locations 1-9 (Fig. 8a). These quartz
veins crosscut the mylonitic fabric but were themselves variably deformed during subsequent shearing, providing direct field evidence that they were broadly coeval with the Woodroffe Thrust, and thus associated with the Petermann Orogeny. Sense of shear is both top-to-north and top-to-south, which is contrary to the dominant top-to-north shear sense associated with the Woodroffe Thrust, but has also been documented by Bell and Johnson (1992) in the Amata region (western edge of Fig. 1). Quartz veins are boudinaged within the mylonitic foliation and, although deformed, did not preferentially localize strain (Figs.
7a,b).

### 6.2 Introduction of carbon

Finely dispersed calcite is locally found with very low modal abundance (typically <1 %) in the otherwise non-carbonaceous rocks of the central Musgrave Block (Fig. 7c). Calcite-bearing samples are present throughout the study area, but are generally more common towards the north (Fig. 8b). Microstructures indicate that the fine-grained (<10 μm) calcite nucleated during
shearing (Fig. 7c). Calcium was made available from recrystallization of plagioclase to new grains with lower anorthite content (Wex et al., 2017), but carbon requires an external fluid since the protoliths are entirely non-carbonaceous. In the attempt to establish the origin of this fluid, carbon and oxygen isotopes of calcite were measured (Supporting Information S2), yielding mean average values of -4.1 ‰ for $\delta^{13}C_{Cal}$ (V-PDB) and +10.1 ‰ for $\delta^{18}O_{Cal}$ (SMOW). Within the same samples, the whole rock isotopic signature $\delta^{13}C_{whole\ rock}$ (V-PDB) is always lower (on average by 2.7 ‰) than the corresponding $\delta^{13}C_{Cal}$ values.





**Figure 7: Field and thin section images providing evidence for the presence or absence of aqueous fluids during deformation. (a) Syntectonically developed quartz veins (encircled) crosscut the mylonitic foliation of the Woodroffe Thrust but were subsequently sheared and rotated, consistent with overall top-to-north shearing. The quartz vein at the bottom of the picture is boudinaged and did not localize deformation. Outcrop NW15-264 (coordinates: 131.46883, -25.83040; location 1 in Fig. 1). (b) Quartz vein adjacent to a sinistral shear zone that is not reactivated, even though shear zone and quartz vein are almost parallel to each other. Abbreviation: $S_m$ = mylonitic foliation. Outcrop SW13-200 (coordinates: 131.73810, -25.99564; location 9 in Fig. 1). (c) Finely dispersed calcite (high birefringence) between recrystallized feldspar grains under crossed-polarized light. Thin section is not oriented. Sample NW14-423A (coordinates: 131.84368, -26.11423; location 10 in Fig. 1). (d) Plagioclase clast with muscovite (higher birefringence) and epidote (lower birefringence) inclusions under crossed-polarized light. Thin section is oriented N-S (left-right). Sample SW14-214A (coordinates: 131.44416, -25.90284; location 5 in Fig. 1). (e) Plagioclase clast with epidote inclusions under crossed-polarized light. Thin section is oriented N-S (left-right). Sample GW13-415 (coordinates: 131.66256, -25.99928; location 8 in Fig. 1). (f) Plagioclase clast with kyanite inclusions (greenish needles) and neo-crystallized garnet under plane-polarized light. Thin section is not oriented. Sample SW13-167 (coordinates: 131.77475, -26.30845; location 14 in Fig. 1). (g) Inclusion-free plagioclase clast under plane-polarized light. Thin section is oriented NNE-SSW (left-right). Sample SW13-318 (coordinates: 132.14311, -25.99142; location 6 in Fig. 1).**

## 6.3 Plagioclase stability and breakdown

Plagioclase recrystallized in the Woodroffe Thrust mylonites and associated shear zones (Bell and Johnson, 1989; Wex et al., 2017) forming typical "core-and-mantle structures". Mineral inclusions within plagioclase clasts are common and allow the distinction of four different types of clasts, respectively termed microstructures 1 to 4:

(1) Plagioclase studded with abundant inclusions of epidote and muscovite (Fig. 7d). This type is, with one exception, restricted to the northern locations 1-9 (Fig. 8c).

(2) Plagioclase containing only epidote inclusions (Fig. 7e), with a modal abundance far lower than that of epidote + muscovite inclusions of microstructure 1. This microstructure is restricted to the central locations 8-13 (Fig. 8c).

(3) Plagioclase crowded with kyanite needle inclusions (Fig. 7f) (see Supporting Information B of Wex et al. (2017) for identification techniques). This microstructure is restricted to the southernmost locations 13-15 (Fig. 8c).

(4) Plagioclase free of inclusions (Fig. 7g). This microstructure is found in the locations 4-13 (Fig. 8c).

Figure 8c shows that the type of inclusions in plagioclase varies in a N-S direction, i.e. parallel to the tectonic transport direction of the Woodroffe Thrust. From north to south, inclusions progressively change from muscovite + epidote (microstructure 1), to epidote (microstructure 2), and to kyanite (microstructure 3), with inclusion-free plagioclase clasts (microstructure 4) occurring throughout. There is no apparent variation of the type of plagioclase inclusions along strike of the Woodroffe Thrust (i.e. E-W).



Figure 8: Regional variation in the development of (a) syntectonic quartz veins, (b) the introduction of carbon and (c) plagioclase
stability and breakdown, each plotted against sample latitude. The regional temperature gradient (Wex et al., 2017) and the position
of the studied locations (Fig. 1) are indicated.





## 7 Abundance of hydrous minerals

The modal abundance of hydrous minerals in deformed and undeformed pseudotachylytes (representative compilation in Supporting Information S3) from felsic footwall and hanging wall samples was determined by image analysis of backscattered electron (BSE) images (method outlined in the Supporting Information S1). The different amounts of hydrous minerals between the pseudotachylytes should reflect compositional variations between the host rocks from which they formed. The results, summarized in Table 4, indicate that matrix mineral assemblages of felsic pseudotachylytes from the hanging wall and footwall are similar to each other. These assemblages dominantly comprise Pl + Kfs + Qz + Bt + Mag + Ilm with individual samples also containing Ep, Grt, Cpx, Opx, Ky, Ms, Rt or Hbl. However, there is a strong contrast with regard to the modal abundance of hydrous minerals. Samples have been sorted into northern (locations 1-5 in Fig. 1), central (locations 6-9 in Fig. 1) and southern location groups (locations 11-14 in Fig. 1), revealing that: (i) the abundance of hydrous minerals decreases from north to south in both the hanging wall and the footwall; and (ii) at similar latitude, the footwall rocks are more hydrous than the hanging wall rocks. Two samples (SW14-029A and SW14-179) do not fit this regional trend and these outliers have been excluded in calculating the mean values, as they mask what are otherwise apparent trends.






**hanging wall of the Woodroffe Thrust**

| location (Fig. 1) | sample | pristine | sheared | BSE images | coordinates (WGS 1984) | mineral assemblage (pseudotachylyte) | hydrous minerals (%) | |
|---|---|---|---|---|---|---|---|---|
| 2 | SW13-174A | x | | 3 | 131.45304 -25.84532 | Pl + Kfs + Bt + Cpx + Qz + Mag + Ilm | 10 | north |
| 3 | NW13-139 | x | | 3 | 131.44826 -25.85366 | Kfs + Pl + Mag + Bt | 4 | north |
| 3 | FW13-096 | x | | 2 | 131.45034 -25.85414 | Pl + Kfs + Bt + Hbl + Ep + Mag | 7 | north |
| 3 | SW13-102 | x | | 4 | 131.45164 -25.85433 | Pl + Bt + Opx + Qz + Mag | 17 | north |
| | | | | | | | av. 10% | |
| 8 | SW13-191 | x | | 4 | 131.66264 -25.99911 | Pl + Cpx + Bt + Qz + Opx + Mag | 16 | central |
| 9 | SW14-029B | | x | 3 | 131.74496 -26.00093 | Pl + Kfs + Bt + Qz + Grt + Ep + Mag | 13 | central |
| 9 | SW14-029A | | x | 3 | 131.74496 -26.00093 | Hbl + Pl + Kfs + Qz + Bt + Ilm | 33* | central |
| 9 | SW14-029A | x | | 2 | 131.74496 -26.00093 | Kfs + Pl + Qz + Bt + Mag | 3 | central |
| | | | | | | | av. 8% [12%] | |
| 9 | SW14-025 | | x | 6 | 131.73269 -26.01569 | Pl + Kfs + Opx + Qz + Bt + Mag + Ilm + Cal | 2 | central |
| 9 | NW13-203 | x | | 3 | 131.74318 -26.00442 | Kfs + Pl + Grt + Bt + Qz + Mag | 5 | central |
| 11 | SW14-181A | x | | 5 | 131.92595 -26.17664 | Kfs + Pl + Qz + Cpx + Grt + Mag + Bt | 1 | south |
| 11 | SW14-181B | x | | - | 131.92595 -26.17664 | Pl + Kfs + Qz + Cpx + Grt + Ilm + Bt | <1 | south |
| | | | | | | | av. 1% | |
| 11 | SW14-181B | x | | 2 | 131.92595 -26.17664 | Pl + Kfs + Grt + Qz + Cpx + Bt + Ilm | 3 | south |
| 11 | SW14-181B | x | | - | 131.92595 -26.17664 | Pl + Kfs + Grt + Opx + Cpx + Qz + Ilm ± Bt | <1 | south |

**footwall of the Woodroffe Thrust**

| location (Fig. 1) | sample | pristine | sheared | BSE images | coordinates (WGS 1984) | mineral assemblage (pseudotachylyte) | hydrous minerals (%) |
|---|---|---|---|---|---|---|---|
| 2 | SW13-171 | x | | 3 | 131.45397 -25.84535 | Qz + Bt + Ms + Ep + Mag + Pl ± Kfs | 40 |
| 2 | SW13-171 | | x | 3 | 131.45397 -25.84535 | Qz + Bt + Ms + Ep + Mag + Pl ± Kfs | 40 |
| 3 | NW13-134A | | x | 2 | 131.45284 -25.85477 | Pl + Qz + Kfs + Ep + Bt | 20 |
| 3 | FW13-093 | | x | 3 | 131.45213 -25.85455 | Pl + Qz + Kfs + Hbl + Bt + Mag + Ilm | 29 |
| 5 | SW14-228A | x | | 3 | 131.44158 -25.90365 | Pl + Kfs + Ep + Bt + Mag + Ilm | 20 |
| | | | | | | | av. 30% |
| 6 | SW13-321 | | x | 5 | 132.14333 -25.99178 | Pl + Kfs + Qz + Bt + Ep + Grt + Mag + Ilm | 13 |
| 6 | SW13-323A | | x | 4 | 132.14344 -25.99211 | Pl + Kfs + Bt + Qz + Mag + Ilm | 10 |
| 6 | SW13-323A | x | | 3 | 132.14344 -25.99211 | Pl + Kfs + Bt + Qz + Mag + Ilm | 10 |
| 6 | SW13-323A | | x | 3 | 132.14344 -25.99211 | Pl + Kfs + Bt + Qz + Mag + Ilm | 8 |
| 8 | SW13-192 | | x | 2 | 131.66258 -25.99915 | Kfs + Pl + Qz + Bt + Ep + Hbl + Grt + Mag | 8 |
| | | | | | | | av. 10% |
| 11 | SW14-179 | x | | 3 | 131.92603 -26.17661 | Pl + Qz + Kfs + Hbl + Grt + Opx + Ilm + Mag ± Bt | 21* |
| 12 | SW13-134 | | x | 2 | 131.87913 -26.21151 | Pl + Kfs + Qz + Bt + Opx + Grt + Ilm | 3 |
| 13 | SW14-237C | x | | 4 | 131.83544 -26.25303 | Kfs + Pl + Bt + Mag | 5 |
| 13 | SW14-244 | | x | 2 | 131.83949 -26.25472 | Pl + Kfs + Qz + Bt + Hbl + Ilm + Grt | 7 |
| 14 | SW13-159 | | x | - | 131.77375 -26.30666 | Pl + Kfs + Cpx + Grt + Qz + Ky + Ilm + Rt | 0 |
| | | | | | | | av. 4% [7%] |

**Table 4: Modal abundance of hydrous minerals in felsic pseudotachylytes from the hanging wall and footwall of the Woodroffe Thrust, central Musgrave Block[a].**

[a] Extreme outliers (*) are not considered. Mineral assemblages are listed in order of decreasing modal abundance (from left to right) with hydrous minerals underlined. A representative compilation of pseudotachylytes is given in the Supporting Information S3.



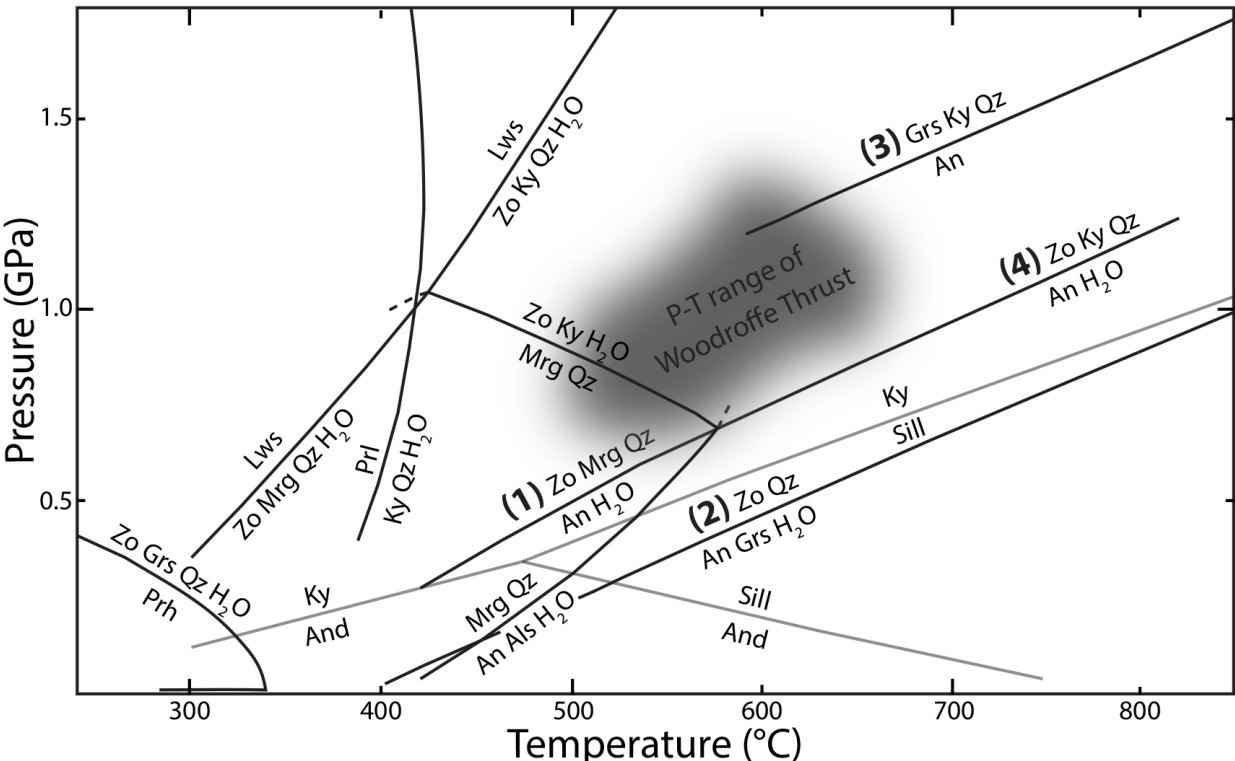

**Figure 9: Petrogenetic grid with some calculated phase relations in the system CaO-Al₂O₃-SiO₂-H₂O (CASH). Note that there is a slight mismatch between plagioclase breakdown reactions as stated in the main text and those highlighted in the diagram, because the components K₂O and Fe₂O₃ are not considered. Consequently, the diagram does not include orthoclase, whereas epidote and muscovite appear as zoisite and margarite, respectively. Data for the P-T range of the Woodroffe Thrust is taken from Wex et al. (2017), while the petrogenetic grid is simplified after Chatterjee et al. (1984), excluding phase relations that are not pertinent to the current study.**

## 8 Discussion

### 8.1 Plagioclase breakdown reactions

Within the range of midcrustal to lower crustal conditions, as estimated for the Woodroffe Thrust (Wex et al., 2017), the following plagioclase breakdown reactions are relevant (Fig. 9):

(1)  An + Or + H₂O = Ms + Ep + Qz (Kretz, 1963; Ramberg, 1949)

(2)  An + Grs + H₂O = Ep + Qz (Kretz, 1963)

(3)  An = Grs + Ky + Qz (Boyd and England, 1961; Hariya and Kennedy, 1968)

(4)  An + H₂O = Ep + Ky + Qz (Goldsmith, 1982; Kretz, 1963; Ramberg, 1949)



For the sake of simplicity, none of these reactions consider the presence of iron. However, iron is certainly necessary to account
for the crystallization of epidote and was potentially derived from relict iron-bearing minerals (e.g., magnetite or ilmenite).
Microstructure 1 is unequivocally correlated with hydration reaction 1. Reaction 2 is capable of producing microstructure 2,
but garnet has been interpreted to serve as a calcium-sink rather than source during the high-pressure Petermann Orogeny
(Camacho et al., 2009). Alternatively, we propose that the decrease in anorthite content during plagioclase recrystallization
(Wex et al., 2017) provided the necessary source of calcium for producing microstructure 2. Microstructure 3 is correlated
with reaction 3. Reactions 3 and 4 represent the high-pressure breakdown of plagioclase under anhydrous and hydrous
conditions, respectively. According to Wayte et al. (1989), the transition between the two competing reactions occurs at a
water activity of ca. 0.004 for pressure and temperature conditions similar to those in the more southerly footwall locations. In
samples with microstructure 3, epidote is never observed as a secondary inclusion phase together with kyanite (Supporting
Information S4), indicating that reaction 3 was always preferred over reaction 4.

### 8.2 Fluid activity

In the southern locations 13-15 (Fig. 1), plagioclase breakdown by reaction 3 indicates that water was not sufficiently available
to drive reaction 4 (Goldsmith, 1980, 1982; Wayte et al., 1989). Consequently, reaction 3 is a good indicator for very low
water activities (<0.004; Wayte et al., 1989) and the absence of a free aqueous fluid phase during deformation, since only very
small amounts of water (ca. 20 ppm) are required for mineral reactions in a solid silicate system (Milke et al., 2013). Similarly,
relict plagioclase clasts without any inclusions (microstructure 4) also indicate the absence of free aqueous fluids, as the studied
rocks were all metastable with respect to hydration reactions 1, 2 and 4 (Fig. 9). Consequently, any of these breakdown
reactions would have rapidly consumed any available free fluid, since all other reactants were present in the studied samples.
Vice versa, plagioclase breakdown by reaction 1 clearly indicates the presence of a free aqueous phase during deformation.
Reaction 2 also involves hydration, as indicated by the marginal presence of epidote in plagioclase clasts (microstructure 2).
However, free aqueous fluids were not sufficiently available to facilitate reaction 1. Hence, we favor the interpretation that
microstructure 2 indicates largely anhydrous conditions, with only very minor fluid introduction.

Based on the classification above, the regional availability of a free aqueous fluid phase during deformation, respectively
termed "wet" and "dry", can be determined from the mineral inclusions in plagioclase (Fig. 8c) and the corresponding inferred
breakdown reactions (Sect. 8.1). The distinction is purely qualitative and refers only to whether or not free aqueous fluid was
sufficiently available to facilitate the hydrous breakdown reaction 1. Consequently, the studied field area is characterized as
dominantly "dry" (microstructures 2, 3 and 4) with a progression towards locally "wet" conditions (microstructure 1) in most
of the northernmost exposures (Fig. 10). The regional variation in (1) the development of syntectonic quartz veins (Fig. 8a)
and (2) the introduction of carbon (Fig. 8b) are each consistent with this interpretation (Fig. 10). However, within a single
location, individual samples can have a "wet" microstructure 1 while others still preserve a "dry" microstructure 4 (Fig. 8c),
indicating that the fluids were only present on a very local scale. This is additionally supported by the coeval development of





a hydrous (Pl + Opx + Grt + Hbl + Ilm + Mag ± Cpx ± Bt) and an anhydrous mineral assemblage (Pl + Opx + Grt + Cpx + Kfs + Qz + Ilm + Mag) within a single thin section of a sheared dolerite dyke from the footwall of the Woodroffe Thrust (Supporting Information S5).


**Figure 10: Projected schematic cross section through the central Musgrave Block. The horizontal scale is compressed by a factor of 3.5 with respect to the vertical scale. The regional temperature gradient is taken from Wex et al. (2017).**



### 8.3 Fluid source

Based on the above quantification of fluid activity in the study area, it is evident that aqueous fluids and $CO_2$-dominated brines were introduced in the northern exposures of the Woodroffe Thrust (Fig. 10). However, there does not seem to be any obvious link between the infiltration of the aqueous fluids and that of the $CO_2$-dominated brines, since samples that preserve a "wet" microstructure 1 are not necessarily calcite-bearing and vice versa (Figs. 8b,c).

### 8.3.1 Aqueous fluids

Aqueous fluids infiltrating into the footwall of the Woodroffe Thrust were unlikely to have been derived from the "dry" hanging wall (Table 4) and consequently, must originate from units within or underlying the current level of exposure of the footwall. Gneisses and granitoids in the "wet" northern part of the study area are clearly interleaved with and juxtaposed onto the basal units of the Amadeus Basin (Wells et al., 1970), as evident from outcrops of Dean Quartzite (Forman, 1965; Young et al., 2002) immediately west of the Kelly Hills klippe (locations 1-5 in Fig. 1). These Neoproterozoic sedimentary rocks

represent an ideal source for aqueous fluids since they were metamorphosed and dehydrated during the Petermann Orogeny (Wells et al., 1970). However, based on regional-scale geometric reconstructions, it has been argued that the Woodroffe Thrust and potentially underlying thrust planes developed in-sequence (Wex et al., 2017). If this is true, then the Dean Quartzite was only imbricated below the northernmost studied locations after movement and deformation on the Woodroffe Thrust had largely ceased and the dynamic microstructures had already been frozen in. Therefore, we prefer a model in which the aqueous

fluids were released from the granitoids and upper amphibolite facies gneisses within the footwall. Such a model is in agreement with the fact that the regional trend towards higher abundance of hydrous minerals in the north parallels the shift towards "wet" conditions, as indicated by the plagioclase breakdown reactions (Fig. 10). An internal source is also consistent with the conclusion that "wet" conditions were only present on a very local scale and that "wet" and "dry" samples are often preserved in close proximity. Consequently, we consider that the studied field area was not infiltrated by externally derived

fluids and remained a relatively closed system.

### 8.3.2 $CO_2$-dominated brines

The fact that the gneisses and granitoids of the central Musgrave Block are entirely non-carbonaceous (Collerson et al., 1972; Major, 1973; Major and Conor, 1993; Scrimgeour and Close, 1999) argues for an external source of the $CO_2$-dominated brines. Carbon and oxygen isotropic signatures were measured in order to provide constraints on its nature. The results for $\delta^{18}O$ are

in agreement with calcite crystallization temperatures of 500-600 °C, while $\delta^{13}C$ values are rock-buffered, but potentially indicate a mantle origin (Supporting Information S2).





### 8.4 Distribution of ductile deformation

The distribution of thorium establishes that the Woodroffe Thrust mylonites preferentially developed in the Mulga Park rather

than the Fregon Subdomain (Fig. 6). This conclusion is in agreement with our field observations and with those of Bell and Etheridge (1976), Camacho et al. (1995) and Flottmann et al. (2004). However, it is evident from our results that in the southern locations (11-14 in Fig. 1) the lowermost hanging wall was also involved in the mylonitization process (Table 3). We discuss below the potential processes guiding the large-scale distribution of ductile deformation in the Woodroffe Thrust.

### 8.4.1 Footwall/hanging wall reworking

The proportion of the total shear strain accommodated in the narrow mylonitic to ultramylonitic zones developed in the

lowermost hanging wall and uppermost footwall cannot be determined. However, what could be determined in this study is the relative thickness of the mylonite zones, using thorium concentrations to distinguish the original hanging wall and footwall protoliths. From this it is established that the hanging wall-derived parts of the Woodroffe Thrust generally make up <10% of the entire width of the mylonitic zone. This preferential development of a broader mylonite zone in the Mulga Park Subdomain footwall rather than in the Fregon Subdomain hanging wall is contrary to the expected simple model of a thrust or reverse fault

system (e.g., Mancktelow, 1985, his Fig. 11). Such a model would predict an asymmetric strain profile where the mylonite zone occurs in the initially "hotter" hanging wall rather than the "colder" footwall. An inverse distribution of ductile deformation, as a consequence of asymmetric thermal weakening, would be in agreement with the flower-like structure model proposed by Camacho and McDougall (2000) for the central Musgrave Block. Based on the preservation of pre-Petermann K-Ar, $^{40}$Ar/$^{39}$Ar and Rb-Sr ages in hornblende, muscovite, biotite and K-feldspar in the undeformed gneissic country rocks,

Camacho and McDougall (2000) argued that the Fregon Subdomain was rapidly buried and exhumed in less than 40 Myr. Consequently, these rocks failed to thermally equilibrate to temperatures above 350 °C at a pressure of ~1.2 GPa. With this tectonic model, the hanging wall should have been at temperatures <350 °C (Camacho and McDougall, 2000) and thus should have been colder than the footwall (>500 °C; Wex et al., 2017). In such a model, thermal weakening could indeed account for preferential mylonitization of the Mulga Park Subdomain. However, Wex et al. (2017) and Hawemann et al. (2017)

demonstrated that Petermann Orogeny metamorphic assemblages throughout the study area are directly comparable in dynamically and statically recrystallized units, arguing that the estimated metamorphic conditions were ambient and differed little between shear zones and country rock. In addition, Wex et al. (2017) found that the syntectonic metamorphic conditions are similar in the hanging wall and footwall at the same locality, arguing against thermal weakening as a cause of the preferential mylonitization of the Mulga Park Subdomain (Camacho and McDougall, 2000).

Pseudotachylytes have been identified as preferred nucleation discontinuities for shearing under mid- to lower crustal conditions (Andersen and Austrheim, 2006; Austrheim and Andersen, 2004; Hawemann et al., 2014; Lund and Austrheim, 2003; Menegon et al., 2017; Passchier, 1982; Pennacchioni and Cesare, 1997; Pittarello et al., 2012; Wex et al., 2014, 2017). In the Musgrave Block however, the presence of precursor pseudotachylyte cannot account for the observed inverse

distribution of ductile deformation, as they are significantly more abundant in the hanging wall compared to the footwall
(Camacho et al., 1995; Lin et al., 2005).

A potential explanation for the observed inverse gradient of ductile deformation along the Woodroffe Thrust is water-assisted
weakening (e.g., Griggs, 1967, 1974; Griggs and Blacic, 1965; Hobbs, 1985; Kronenberg et al., 1990; Stünitz et al., 2017;
Tullis and Yund, 1989). As discussed in Sect. 8.3.1, the rocks in the study area represent a relatively closed system with respect
to the presence or absence of aqueous fluids. Hence, the availability of aqueous fluids is directly linked to the abundance of
hydrous minerals in these rocks, which is observed to decrease from north to south (Fig. 10). Similarly, the contrasting
abundance of hydrous minerals in hanging wall and footwall (Table 4) indicates that there is consistently a higher potential for
generating aqueous fluids in the footwall than in the hanging wall. Overall, the majority of the studied rocks are relatively
"dry", so that only slight differences in the abundance of water could facilitate an asymmetric localization of ductile
deformation as a consequence of water-assisted weakening. The preferential mylonitization of the Mulga Park relative to the
Fregon Subdomain (Fig. 6) therefore seems to be largely controlled by the precursor mineralogy established as the result of
the earlier (ca. 1200 Ma) Musgravian Orogeny metamorphism, with peak conditions of upper amphibolite facies in the footwall
but granulite facies in the hanging wall. This supports the initial hypothesis of Bell and Etheridge (1976) and Camacho et al.
(1995) that a more hydrous footwall underlying an anhydrous hanging wall can facilitate an inverted gradient of ductile
deformation.

These arguments may explain the overall preferential localization of deformation in the Mulga Park Subdomain, but fail to
provide an adequate explanation for why the lowermost Fregon Subdomain was also incorporated into the mylonites (Figs. 2,
6, 10). The generally low abundance of hydrous minerals in both hanging wall and footwall in the "dry" southern locations
(11-14 in Table 4) potentially promoted a similar rheological response in both units. Locally, there may be no contrast at all in
protolith composition, which might explain the marginal mylonitization of the lowermost Fregon Subdomain. Reworking of
the hanging wall may also have been guided by the presence of pseudotachylytes, which are ubiquitous and voluminous in the
immediate hanging wall of the Woodroffe Thrust (Camacho et al., 1995; Lin et al., 2005).

**8.4.2 Variation in mylonite thickness**

Mylonite thickness in the study area does not appear to significantly vary parallel to strike (i.e. E-W). However, from south to
north, i.e. parallel to the direction of tectonic transport, the Woodroffe Thrust mylonitic zone gradually decreases in thickness
from over 600 m to less than 100 m (Table 1), indicating a 6-fold increase in shear strain (and therefore average strain rate)
within the mylonites, assuming that the relative displacement across the Woodroffe Thrust was constant along the entire ca.
60 km transect. This trend from south to north is accompanied by a slight decrease in metamorphic temperature of ca. 100 °C
(Wex et al., 2017) and the shift from "dry" to "wet" conditions, as reflected in the increasing abundance of hydrous minerals
(Fig. 10). The thickness of large-scale shear zones is considered to decrease with decreasing temperature and depth (Platt and





Behr, 2011a, 2011b) and displacement (Hull, 1988), but may also decrease with increasing fluid-rock interaction as a
consequence of volume loss (e.g., Newman and Mitra, 1993). A point in the footwall only enters the shear zone when it passes
the toe of the thrust (that is where the thrust meets the surface). This could potentially lead to a variation in the finite shear
strain experienced in the footwall. However, the whole of the exposed ca. 60 km N-S section was formerly at mid- to lower
crustal level (Wex et al., 2017) and thus nowhere near the thrust toe. There is also currently no evidence for major splays of

the Woodroffe Thrust into the hanging wall. We therefore assume that the studied section experienced more or less the same
relative displacement and that variation in this parameter cannot account for the difference in mylonite thickness. It was argued
above that the greater abundance of hydrous minerals in the footwall Mulga Park Subdomain compared to the hanging wall
Fregon Subdomain, reflecting peak metamorphic conditions of upper amphibolite facies and granulite facies respectively
during the earlier Musgravian Orogeny, resulted in slightly "wetter" conditions in the footwall during activity of the Woodroffe

Thrust. This was proposed as an explanation for the broader zone of mylonites in the footwall, which in turn would argue
against the increasingly hydrous conditions toward the north as an explanation for the decrease in overall mylonite width. The
general decrease in mylonite zone thickness toward the north is therefore interpreted to be due to the established decrease in
metamorphic temperature, promoting localization in a narrower zone.

## 9 Conclusions

Field and thin section observations establish that the rocks of the central Musgrave Block were predominantly "dry" during
development of the midcrustal Woodroffe Thrust during the ca. 560-520 Ma Petermann Orogeny, but with a progression in
the thrust direction towards locally "wet" conditions in some of the northernmost exposures. This is indicated by: (1) rare
occurrence of syntectonic quartz veins and quartz-rich pegmatites (locally found only in the north), (2) metastability of
plagioclase in the presence of K-feldspar, which rarely shows significant sericitization via the reaction $An + Or + H_2O = Ms$

$+ Ep + Qz$ (more common towards the north) and (3) preferential high-pressure breakdown of plagioclase via the reaction $An
= Grs + Ky + Qz$ (common in the southerly exposures), rather than $An + H_2O = Ep + Ky + Qz$. Aqueous fluids were most
likely derived internally from hydrous minerals within the footwall gneisses and granitoids, implying that the rocks in the
study area were a relatively closed system.

The thickness of the Woodroffe Thrust mylonites generally increases with increasing metamorphic grade and does not appear

to be linked to the presence or absence of an aqueous fluid. However, atypical of a thrust, ductile deformation is more
extensively developed in the footwall rocks and only marginally involved several tens of metres of the lowermost hanging
wall. The inverse gradient of ductile deformation cannot be explained by thermal weakening or the distributed presence of
pseudotachylyte (acting as preferred nucleation sites for shearing), but rather by preferential water-assisted weakening in the
"wetter" footwall compared to the "dry" hanging wall. This reflects the earlier (Musgravian Orogeny) peak metamorphic

conditions (granulite facies in the hanging wall and upper amphibolite facies in the footwall) and the contrasting availability
of aqueous fluids derived from relict hydrous minerals in the footwall and hanging wall.



## Data availability

Supplementary data are available in the Supporting Information "S1" to "S5" and further information can be obtained on request from the corresponding author.

## Competing interests

The authors declare that they have no conflict of interest.

## Acknowledgments

We thank the communities of the Anangu Pitjantjatjara Yankunytjatjara Lands (APY) for granting us access to the Musgrave Ranges. Logistical support from the Northern Territory Geological Survey (NTGS) of Australia, Prof. B. Tikoff (Univ. 490 Wisconsin, Madison) and Shane and Alethea Nicolle are gratefully recognized. We further acknowledge the support of Dr. K. Kunze from the Scientific Center for Optical and Electron Microscopy (ScopeM) at the ETH, Zurich. Dr. J. Eikenberg is thanked for supervising and conducting the thorium measurements at the Paul Scherrer Institute (PSI) in Villigen, Switzerland. We further acknowledge the support of M. Jaggi who carried out the stable isotope analyses at the Geological Institute at ETH, Zurich. This project was financed by the Swiss National Science Foundation (SNF) Grant 200021_146745 and by the 495 University of Padova (BIRD175145/17: The geological record of deep earthquakes: the association pseudotachylyte-mylonite).

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
