# Peer review of "Inverted distribution of ductile deformation in the relatively "dry" middle crust across the Woodroffe Thrust, central Australia"

_Solid Earth, 2018_

## Short Comment (SC1) · 16 Mar 2018

This manuscript presents a variety of field, chemical, and microstructural observations and data related to the Woodroffe Thrust in central Australia, with the goal to better understand how the mylonite zone evolved, and in particular, why the majority of the deformation is developed in the footwall rather than in the hangingwall. This is an interesting and well-conceived contribution about deep-crustal shear zone deformation, and the topic is of broad interest to an international audience. I enjoyed reading it.

The overall interpretation appears to be sound and based on a broad range of observation types. However, there are two main issues that I would suggest the authors

consider carefully:

The first has to do with organization and lack of necessary background information. Much of the necessary information appears to have been published already in Wex et al. (2017, Tectonics), but more than citations to that work need to be presented here. For example, there is a distinct need for some description of the "starting material" that was reworked in this shear zone, or at least some description of representative lithologies, assemblages, and estimated equilibrium conditions for those starting assemblages (presumably Musgravian). This has important implication for the authors' interpretation that the fluid source was the lower plate and that the shear zone system was essentially "closed." The lower plate rock is currently described as "upper amphibolite-facies during Musgravian without further details. But were these rocks reworked by the Woodroffe deformation in a retrograde or prograde sense? If prograde, then they could conceivably produce free fluid internally through dehydration reactions. But if in a retrograde sense, then they would have consumed water, if present, rather than produce it. The authors could call upon dehydration-driven fluids from deeper levels in the footwall, but then it is not a "closed system." The whole calcite and O and C isotope story seems to point to an external source of fluid too. Some clarification is needed, and probably a summarization of what Wex et al. (2017) learned about these rocks in the previous study would help a lot. This suggestion extends to the pseudotachylyte system as well, which is used in the current study but with very little description what their role in the evolution of the shearing history is. The authors mention evidence for two stages of deformation at different conditions in the shear zone and give P-T conditions, but does that really reflect distinctly different time periods? What is the basis for "regional temperature gradient" that is shown in Fig. 8 and 10??? There is no description of where that came from other than a citation to Wex et al. (2017), but it basis is pretty critical this proposed story and interpretations here too.

The second is also primarily an organization thing and has to do with sample localities, the various geographic groupings, and the N-S transect. This is all quite confusing.

I suggest the authors start early in the manuscript with a description of the sampling strategy as essentially on a N-S transect like that shown at the bottom of the current Fig. 8 – that is easy to see and easy to keep referring to, but show it in a simplified form earlier than in the 8th figure. Right now, the fact that sample localities are broken into various northern and southern plus/minus central groups based on different datasets is convoluted. For the bulk Th measurements, "northern" is locations 1 through 6. According to Fig. 8, station 7 is plots more northerly than 6 so you really should include 7 too. But for plagioclase stability, "northern" is stations 1 through 9. And for abundance of hydrous minerals, "northern" is only stations 1 through 5. Stop all of the group attempts and simply show how the various observations and data change along the transect – the trends will be the same and much easier to follow.

The final major point is that the beginning of the discussion on feldspar breakdown reactions and fluid activity in sections 8.1 and 8.2 is vague and not particularly strong. How can reactions 1 and 2 be pitted against one another when they reflect different chemical systems? One has K in it and the other does not. There is another vague reference to Wex et al. (2017) for plagioclase chemical variability during recrystallization (dynamic?) but no details – this information is important to the discussion and should be explained in more detail, either here or earlier in the manuscript in a summary of the earlier study. There is a claim of "coeval development" of a (synkinematic?) dry assemblage and a wet one in the same rock with reference to S5. But the figure in S5 only shows a dry assemblage, and it is a static texture (could that be Musgravian even?). The claim that fluid activity has been "quantified" is not justified; this is a qualitative evaluation, not a quantitative one. However, there is still a convincing case that there was indeed more water in the north than in the south; but the case is currently overstated in terms of what has really been constrained petrologically. And Fig. 9 is not necessary.

Other more specific points: Lines 75-80: What is the significance of these earlier mylonites? Lines 90-95: Two stages of deformation and P-T conditions are given, but only

one long set of "stable" mineral assemblages is given. Stable with respect to which stage? Section 5.1: Can an estimated error be quoted for the Th measurements? Line 187: switch 6 ppm and 8 ppm Fig. 4: how is the lower bound of the mylonite zone defined and how well is that constrained? Fig. 5: Airborne Th maps – how sensitive is this measurement to depth? What is the thickness of the Kelly Hills klippe? Could that be a contributor to increased Th signal? Line 207: Why assume all PST is in the hangingwall? Section 6.2: indicate whether these measurements are in interpreted hangingwall or footwall Section 7: why do this only on the PST? Why not directly on the host rocks? Fig. 9 – not needed Line 320: Na is not considered either. Iron could also come from Bt or garnet. Fig. 8 and 10 – where does the regional temperature gradient come from and what is its interpreted meaning? Line 360 – did not quantify water activity Section 8.3.2 – this all supports an external fluid source; so how to reconcile? Line 422-23: distinguish "water weakening" from simply a rheologically weaker assemblage (e.g., higher mica abundance?) Line 437: this also argues against a "closed system" Line 450-451: clarify why increasing fluid-rock interaction would result in volume loss?

-K.H.Mahan

---

## Referee Comment (RC1) · F. Fusseis (Referee) · 19 Mar 2018

Review of Wex et al "Inverted distribution of ductile deformation in the relatively dry middle crust across the Woodroffe Thrust, central Australia"

The authors present a further contribution on this spectacular thrust system, exploring the apparent asymmetry of the shear zone width with respect to the incorporation of the foot- and hanging wall rocks. They confirm that the inverted scenario that favoured mylonitization of the footwall over those of the hanging wall rocks is due to a slightly increased water content in the former, especially in the northern parts of the explored thrust system.

The paper is very well written, beautifully illustrated and a captivating read even late at night. Clearly the paper is companion to other papers from the group, in particular of course the recent work of Wex et al (2017), and I agree with Kevin Mahan that certain shortcuts that arose from this should be corrected to improve the readability of the presented manuscript.

Two aspects struck me as "loose ends" though:

First, I find it curious that the authors miss the opportunity to discuss the significant increase in shear strain due to the narrowing of the Woodroffe thrust from south to north in light of the strain softening mechanisms they envisage. The relatively lapidary last sentence of the discussion (line 463) does, in my eyes, not explain why the Woodroffe thrust is six-fold narrower in the north, especially if the wetter conditions do not account for the pronounced strain localization there. The observation that the thrust narrows so dramatically would invite a more detailed discussion of the microstructural developments along the strain gradients in the north and south, or, in other words, the strain-dependent evolution of the mylonites. Which mechanisms accommodate strain softening in the north and south? Is it possible that the shear zone progressively narrowed during shearing, in analogy with Means' (1995) type 2 shear zones? If so, does this apply to the entire Woodroffe thrust? And if not, why not? I would invite the authors to include a more detailed microstructural description of the evolution from host rock to ultramylonite in both, northern and southern sections, and then integrate these into the discussion of their findings in light of the questions above.

Second, the mantle source of the CO2-dominated brines. If these are indeed mantlederived, how did they migrate along the shear zone? Was there some form of synkinematic porosity? If there was a fluid migrating along the Woodroffe thrust, what was its micromechanical effect in both, the northern and southern sections?

With respect to the determination of the modal abundance of the hydrous minerals (S1.3) – global thresholding on the basis of grey value histograms is rather primitive and

prone to substantial errors – Fiji/ImageJ offers much more sophisticated segmentation algorithms, in particular trainable WEKA segmentation, a machine learning toolbox.

**Florian Fusseis**

СЗ

---

## Author Response (AR1)

**Author comments (AC) and manuscript changes (MC) on interactive comment by K. Mahan**

(kevin.mahan@colorado.edu)

**1)** *The first has to do with organization and lack of necessary background information. Much of the necessary information appears to have been published already in Wex et al. (2017, Tectonics), but more than citations to that work need to be presented here. For example, there is a distinct need for some description of the "starting material" that was reworked in this shear zone, or at least some description of representative lithologies, assemblages, and estimated equilibrium conditions for those starting assemblages (presumably Musgravian). This has important implication for the authors' interpretation that the fluid source was the lower plate and that the shear zone system was essentially "closed." The lower plate rock is currently described as "upper amphibolite-facies during Musgravian without further details. But were these rocks reworked by the Woodroffe deformation in a retrograde or prograde sense? If prograde, then they could conceivably produce free fluid internally through dehydration reactions. But if in a retrograde sense, then they would have consumed water, if present, rather than produce it.*

**AC/MC:** Our geological introduction was indeed very concise. Following the reviewer's suggestion, we have extended the geological introduction of the manuscript to provide additional information regarding the Musgravian Orogeny "starting material" and the estimated metamorphic conditions, also in comparison to the metamorphic overprint during the Petermann Orogeny.

—————————

**2)** *The authors could call upon dehydration-driven fluids from deeper levels in the footwall, but then it is not a "closed system." The whole calcite and O and C isotope story seems to point to an external source of fluid too.*

**AC:** Carbon indeed cannot have an immediately local source, because the studied rocks are initially entirely non-carbonaceous. With regard to carbon the system is clearly not closed. We tried to further constrain the origin using stable isotopes. Unfortunately, the whole calcite isotope story is not conclusive, and our current conclusions based on this data therefore rather speculative. However, our observations and data clearly indicate that the source of the hydrous fluids and that of carbon are not the same. The external carbon source does therefore not provide any constraints for the source of the hydrous fluids. We do not argue that dehydration-driven fluids originate from deep levels in the footwall, but immediately from within the local, exposed footwall units. This would represent a closed system with regard to the aqueous fluids.

**MC:** The interpretation of the calcite isotopic data, has been formulated into more restrained statements (see comment 25) and moved to the appendix in the revised version.

—————————

**3)** *Some clarification is needed, and probably a summarization of what Wex et al. (2017) learned about these rocks in the previous study would help a lot.*

**AC:** Where necessary, Wex et al. (2017) is cited and, where appropriate, we now provide additional information, e.g. that P-T conditions were determined via conventional geothermobarometry rather than just stating the derived numbers. We have avoided providing an extensive and detailed report of the results previously published in Wex et al. (2017), as that would result in unnecessary lengthening of the manuscript. The current manuscript was submitted subsequently to the published companion paper of Wex et al. (2017) in order to allow a systematic development and avoid repetition of previously published data.

**MC:** Additional information on how Wex et al. (2017) constrained the syn-kinematic P-T conditions and the interpreted meaning of these values has been added.

—————————

**4)** *This suggestion extends to the pseudotachylyte system as well, which is used in the current study but with very little description what their role in the evolution of the shearing history is.*

**AC:** The pseudotachylyte system and associated microstructures are only of marginal relevance to the current paper, which is why they were not presented in full detail. These aspects will be thoroughly presented in a follow-up companion paper, which particularly focuses on the brittle-ductile interplay (pseudotachylytes vs. mylonites) and pseudotachylyte development at mid-crustal depths. The proposed manuscript has already been compiled and is ready for submission once the current manuscript is published. The current manuscript will provide some relevant background information for this forthcoming paper, as the paper by Wex et al. (2017) has provided necessary background information for the current submission.
* * *
**5)** *The authors mention evidence for two stages of deformation at different conditions in the shear zone and give P-T conditions, but does that really reflect distinctly different time periods?*

**AC:** The mentioned two stages of deformation are differentiated on the basis of microstructure and petrology and interpreted to represent different time periods during shearing along the Woodroffe Thrust, as discussed in detail in Wex et al (2017). We agree that mentioning this aspect might be confusing to the reader, because it is not crucial in the current manuscript.

**MC:** Since this information is not relevant to the current manuscript we have omitted those parts making reference to the presence of two different stages of deformation.

**6)** *What is the basis for "regional temperature gradient" that is shown in Fig. 8 and 10??? There is no description of where that came from other than a citation to Wex et al. (2017), but it basis is pretty critical this proposed story and interpretations here too.*

**AC:** As discussed in comment 3), we have now added the information that P-T conditions were constrained from conventional geothermobarometry. We provide this information where we first referenced Wex et al. (2017) in reference to the estimated P-T conditions. Then, for subsequent references to the temperature gradient and to the P-T estimates (for instance in Fig. 8 and 10), a simple reference to Wex et al. (2017) should be sufficient.

**MC:** Additional information on how Wex et al. (2017) constrained the syn-kinematic P-T conditions and the interpreted meaning of these values has been added.
* * *
**7)** *I suggest the authors start early in the manuscript with a description of the sampling strategy as essentially on a N-S transect like that shown at the bottom of the current Fig. 8 – that is easy to see and easy to keep referring to, but show it in a simplified form earlier than in the 8th figure.*

**AC:** The locations of all samples are clearly given in Figure 1, the very first figure of the paper.

**MC:** In section 3, we now state explicitly that samples were collected "along a N-S traverse, parallel to the direction of thrusting".
* * *
**8)** *Right now, the fact that sample localities are broken into various northern and southern plus/minus central groups based on different datasets is convoluted. For the bulk Th measurements, "northern" is locations 1 through 6. According to Fig. 8, station 7 is plots more northerly than 6 so you really should include 7 too. But for plagioclase stability, "northern" is stations 1 through 9. And for abundance of hydrous minerals, "northern" is only stations 1 through 5. Stop all of the group attempts and simply show how the various observations and data change along the transect – the trends will be the same and much easier to follow.*

**AC:** We agree that this is aspect can cause confusion to the reader.

**MC:** We followed the reviewer's suggestion and stopped our attempts of grouping locations. We now only refer to trends.

———————————

**9)** *The final major point is that the beginning of the discussion on feldspar breakdown reactions and fluid activity in sections 8.1 and 8.2 is vague and not particularly strong. How can reactions 1 and 2 be pitted against one another when they reflect different chemical systems? One has K in it and the other does not. There is another vague reference to Wex et al. (2017) for plagioclase chemical variability during recrystallization (dynamic?) but no details – this information is important to the discussion and should be explained in more detail, either here or earlier in the manuscript in a summary of the earlier study.*

**AC:** We only pit reactions 3 and 4 against each other as these are competing reactions. Reactions 1 and 2 are not meant to be pitted against each other, but to explain the different microstructures 1 and 2, which potentially arise due to the different availability of K.

**MC:** As additional information, we now state the composition of recrystallized feldspar and also the fact that recrystallization was dynamic.

———————————

**10)** *There is a claim of "coeval development" of a (synkinematic?) dry assemblage and a wet one in the same rock with reference to S5. But the figure in S5 only shows a dry assemblage, and it is a static texture (could that be Musgravian even?).*

**AC:** The static microstructural overprint in S5 is consistent with the observations made in other deformed and undeformed samples which have been overprinted during the Petermann Orogeny. We agree that the appendix figure was actually more confusing than helpful to the reader, which is why we dropped it.

**MC:** Appendix S5 and its respective reference in the manuscript have been omitted.

———————————

**11)** *The claim that fluid activity has been "quantified" is not justified; this is a qualitative evaluation, not a quantitative one. However, there is still a convincing case that there was indeed more water in the north than in the south; but the case is currently overstated in terms of what has really been constrained petrologically.*

**AC/MC:** A valid point: we have replaced "quantified" with "estimated", reflecting the fact that the estimations are only qualitative.

———————————

**12)** *Lines 75-80: What is the significance of these earlier mylonites?*

**AC:** We included these earlier mylonites for the sake of keeping the geological history complete. However, we agree that this is only confusing to the reader. The mylonites are described in more detail in Wex et al. (2017), but are not relevant to the current manuscript.

**MC:** The mentioning of the earlier mylonites has been omitted from the manuscript, because they are not relevant here and were not further discussed.

———————————

**13)** *Lines 90-95: Two stages of deformation and P-T conditions are given, but only one long set of "stable" mineral assemblages is given. Stable with respect to which stage?*

**AC:** We agree that this aspect is confusing to the reader. As discussed in comment 5), the mentioned two stages of deformation are irrelevant to the current paper, which is why only a single list of minerals, those stable during the Petermann Orogeny, is given. Since the mentioning of the two stages has now been dropped, a single list of minerals is sufficient.

**14)** *Section 5.1: Can an estimated error be quoted for the Th measurements?*

**AC/MC:** Measurements were run until the error was < 10% (stated in appendix S1). We now provide this information also in the manuscript.
* * *
**15)** *Line 187: switch 6 ppm and 8 ppm*

**AC/MC:** done.
* * *
**16)** *Fig. 4: how is the lower bound of the mylonite zone defined and how well is that constrained?*

**AC:** The lower bound of the mylonite zone is defined by the initial appearance of Petermann mylonitic foliation well characterized by the trend of the stretching lineation and by the top-to-north kinematic indicators. Due to the gradual and irregular nature of the mylonite zone, with high-strain shear zones surrounding less to little deformed low-strain domains on the metre- to kilometre-scale, the lower bound is less well defined than its upper counterpart, but on the whole still reliable.

MC: The information that the lower bound is defined by the initial appearance of mylonites with Petermann kinematics is now provided.
* * *
**17)** *Fig. 5: Airborne Th maps – how sensitive is this measurement to depth? What is the thickness of the Kelly Hills klippe? Could that be a contributor to increased Th signal?*

**AC:** As stated by Jones and Schreib (2007): "The gamma ray signal for natural radioisotopes in rocks comes almost entirely from the top 35 cm (IAEA, 2003)", thus making the thickness of the Kelly Hilly klippe irrelevant to the increased Th signal.

D G JONES AND C SCHEIB. 2007. A preliminary interpretation of Tellus airborne radiometric data. British Geological Survey Commissioned Report, CR/07/061. 70pp.

IAEA. 2003. Guidelines for radioelement mapping using gamma ray spectrometer data. International Atomic Energy Agency IAEA-TECDOC, 1363, pp. 173.

**MC:** This clarification, together with the references, has now been added.
* * *
**18)** *Line 207: Why assume all PST is in the hanging wall?*

**AC:** We did not want to give the impression that pseudotachylytes were exclusively restricted to the hanging wall, since this is clearly not true. However, it is evident from field observations that the largely unsheared pseudotachylyte breccias in the hanging wall are locally sheared and dragged into the Woodroffe Thrust mylonitic zone.

**MC:** The statement has been reformulated into: "Similar field relationships, such as progressive downwards mylonitization of units clearly forming part of the hanging wall, also indicates limited reworking of the Fregon Subdomain at locations 4 and 6 (Fig. 5)".
* * *
**19)** *Section 6.2: indicate whether these measurements are in interpreted hanging wall or footwall*

**AC/MC:** This information is now provided in the Appendix Table S1.
* * *
**20)** *Section 7: why do this only on the PST? Why not directly on the host rocks?*

**AC:** Pseudotachylytes have been identified as preferred discontinuities for nucleation of shear zones under mid- to lower crustal conditions.

**MC:** This concept is now included at the beginning of section 7 with an extensive list of supporting references.
* * *
**21)** *Fig. 9 – not needed*

**AC/MC:** Fig. 9 has been deleted.
* * *
**22)** *Line 320: Na is not considered either. Iron could also come from Bt or garnet.*

**AC/MC:** The equations presented are indeed very simplified but are only taken as indicative of the reactions involved. Biotite and garnet have been added as potential sources of iron.
* * *
**23)** *Fig. 8 and 10 – where does the regional temperature gradient come from and what is its interpreted meaning?*

**AC:** This aspect is discussed in comments 3) and 6).

**MC:** Additional information on how Wex et al. (2017) constrained the syn-kinematic P-T conditions and the interpreted meaning of these values has been added.
* * *
*24) Line 360 – did not quantify water activity*

**AC:** We have replaced "quantified" with "estimated".
* * *
**25)** *Section 8.3.2 – this all supports an external fluid source; so how to reconcile?*

**AC:** Our data does not allow us to reconcile towards a single fluid source. Our observations and results indicate that the $H_2O$ and $CO_2$ each originated from different sources. We argue in favour of an internal source for the hydrous fluids, whereas the fact that we observe syn-kinematic calcite growth in otherwise non-carbonaceous rocks clearly indicates that carbon was introduced externally. Additional stable isotopic analysis on calcite, unfortunately, did not provide any significant constraints, allowing no further conclusions to be drawn without speculation.

**MC:** The interpretation of the isotopic analysis have been shifted entirely into the appendix with only a short summary of the isotopic results provided in the main manuscript. The potential mantle source as the most likely source has been dropped. The different possibilities are still presented in the appendix, however we reconcile by not trying to over interpret our isotopic results, since they are simply not conclusive enough.
* * *
**26)** *Line 422-23: distinguish "water weakening" from simply a rheologically weaker assemblage (e.g., higher mica abundance?)*

**AC/MC:** done.
* * *
**27)** *Line 437: this also argues against a "closed system"*

**AC:** We agree that our argumentation, that the generally low abundance of hydrous minerals in both hanging wall and footwall in the "dry" southern locations potentially promoted a similar rheological response in both units, is indeed rather speculative. It is, in fact, also not supported by our Table 4.

**MC:** We decided to drop the mentioned statement and now solely argue that the local reworking of the lowermost hanging wall was guided by the presence of pseudotachylyte.
* * *
**28)** *Line 450-451: clarify why increasing fluid-rock interaction would result in volume loss?*

**AC:** We admit that there is no evidence for volume loss associated with fluid-rock interaction.

**MC:** The statement is omitted in the new manuscript version.
* * *
**Author Comment (AC) and manuscript changes (MC) on interactive comment by F. Fusseis**

(florian.fusseis@ed.ac.uk)

**1)** *First, I find it curious that the authors miss the opportunity to discuss the significant increase in shear strain due to the narrowing of the Woodroffe thrust from south to north in light of the strain softening mechanisms they envisage. The relatively lapidary last sentence of the discussion (line 463) does, in my eyes, not explain why the Woodroffe thrust is six-fold narrower in the north, especially if the wetter conditions do not account for the pronounced strain localization there. The observation that the thrust narrows so dramatically would invite a more detailed discussion of the microstructural developments along the strain gradients in the north and south, or, in other words, the strain-dependent evolution of the mylonites. Which mechanisms accommodate strain softening in the north and south? Is it possible that the shear zone progressively narrowed during shearing, in analogy with Means' (1995) type 2 shear zones? If so, does this apply to the entire Woodroffe thrust? And if not, why not? I would invite the authors to include a more detailed microstructural description of the evolution from host rock to ultramylonite in both, northern and southern sections, and then integrate these into the discussion of their findings in light of the questions above.*

**AC/MC:** After reconsideration, we agree with the reviewer that our previous conclusion that the narrowing is exclusively controlled by temperature is probably too restrictive and not justified. The last sentence of the discussion has therefore been extended into a more detailed discussion on the potential role of aqueous fluids as an additional or alternative explanation for the narrowing of the Woodroffe Thrust from south to north. We have updated the discussion to argue that an increase in the water content to the north would potentially result in an increase of the effective viscosity ratio between footwall and hanging wall, potential causing a stronger localization toward the interface and a narrower mylonite zone that extends less into the stronger material. Indeed, we envision a progressive narrowing of the Woodroffe Thrust with time, in analogy with Means' (1995) type 2 shear zones. Microstructural and petrological evidence for this is found in the southern outcrops of the Woodroffe Thrust, with the temporal development in detail discussed by Wex et al. (2017). A detailed discussion of the microstructural developments along the strain gradients in the north and south, as requested by the reviewer, has been carried out in the overall framework of our study of the Woodroffe Thrust, but has been drafted into a follow-up companion paper, which focuses particularly on the microstructural gradients parallel to the thrusting direction and the inferred deformation mechanisms in quartz and feldspar, as characteristic for middle to lower continental crust.
* * *
**2)** *Second, the mantle source of the CO2-dominated brines. If these are indeed mantle-derived, how did they migrate along the shear zone? Was there some form of synkinematic porosity? If there was a fluid migrating along the Woodroffe thrust, what was its micromechanical effect in both, the northern and southern sections?*

**AC/MC:** There is no indication that there is preferentially more calcite in the more strongly mylonitic to ultramylonitic rocks, arguing against channeling of the $CO_2$-rich fluids along the Woodroffe Thrust. Furthermore, we agree that the whole calcite isotope story is not conclusive, and our current conclusions based on this data slightly overambitious. The interpretation of the calcite isotopic data has been formulated into more restrained statements and is, in the revised manuscript, restricted to the appendix. A short summary of the isotopic results is still provided in the main manuscript.
* * *
**3)** *With respect to the determination of the modal abundance of the hydrous minerals (S1.3) – global thresholding on the basis of grey value histograms is rather primitive and prone to substantial errors – Fiji/ImageJ offers much more sophisticated segmentation algorithms, in particular trainable WEKA segmentation, a machine learning toolbox.*

**AC:** The FiJi/ImageJ software does provide more sophisticated segmentation tools but it also requires careful checking of the output results. We put particular care in checking that the greyscale histograms

of the collected SEM images allowed a clear separation between the hydrous minerals and the anhydrous minerals. We believe that the determined modal abundance of hydrous minerals with our thresholding technique is quite reliable.

[revised manuscript text omitted]

$\delta^{13}C_{Cal}$ should provide information on the fluid source, since there are no other carbon-bearing phases in the studied rocks (Collerson et al., 1972; Major, 1973; Major and Conor, 1993; Scrimgeour and Close, 1999). $\delta^{18}O_{Cal}$,

95   on the other hand, should reflect the calcite crystallization temperatures. Our results are in agreement with the range of $\delta^{18}O$ values for metamorphic waters (Taylor, 1997, his Fig. 6.4) and carbonates (Coplen et al., 2002, their Fig. 6) at temperatures of 500-600 °C, considering that the corresponding fluid phase should isotopically have been ca. 5-6 ‰ heavier than the crystallizing calcite (Chacko et al., 1991). Our $\delta^{13}C_{Cal}$ results strongly overlap with the typical range of values for a rock-buffered system within igneous and metamorphic rocks (Coplen et al.,

100   2002, their Fig. 4). However, from Table S1 it is evident that $\delta^{13}C_{Cal}$ is never identical to $\delta^{13}C_{whole\ rock}$. Consequently, full rock-buffering has not been achieved and it should be possible to place some constraints on the potential carbon source. The range of determined $\delta^{13}C_{Cal}$ values excludes a marine origin (Coplen et al., 2002, their Fig. 4), but is in agreement with a mantle (Deines, 2002; Javoy et al., 1986) or sedimentary source (Bitter Springs Formation; Hill et al., 2000). Within the underlying Amadeus Basin sediments, the most likely source

105   would have been either the carbonate-bearing Bitter Springs Formation or its lateral equivalent the Pinyinna Beds (Wells et al., 1970; Young et al., 2002). However, both units do not crop out in the study area and, as outlined in Sect. 8.3.1 of the main paper, the Amadeus Basin sediments were potentially only imbricated below the Kelly Hills klippe after shearing on the Woodroffe Thrust had largely ceased.

[Figure]

**Figure S4: Representative BSE images of felsic pseudotachylytes from the hanging wall and footwall of the Woodroffe Thrust arranged into northern** **, central**  **and southern location groups (**see Table 4 **of the main paper). Overview thin section images are roughly 4 cm in width. (top left) Unoriented sample FW13-096 (coordinates: 131.45034, -25.85414; location 3 in Fig. 1 of the main paper). (top right) Unoriented sample FW13-093 (coordinates: 131.45213, -25.85455; location 3 in Fig. 1 of the main paper). (centre left) Unoriented sample SW14-029A (coordinates: 131.74496, -26.00093; location 9 in Fig. 1 of the main paper). (centre right) Sample SW13-321 (coordinates: 132.14333, -25.99178; location 6 in Fig. 1 of the main paper). Thin section is oriented N-S (left-right). (bottom left) Loose sample SW14-181B (coordinates: 131.92595,**

-26.17664; location 11 in Fig. 1 of the main paper). (bottom right) Unoriented sample SW14-237C (coordinates: 131.83544, -26.25303; location 13 in Fig. 1 of the main paper).

**S4 Plagioclase stability microstructure 3**

[Figure]

125    **Figure S5: BSE image of plagioclase with kyanite inclusions (microstructure 3). Elemental maps (Al, Ca, Fe) do not indicate the presence of epidote. The host rock is a statically overprinted dolerite dyke with a stable mineral assemblage of Pl + Cpx + Grt + Ilm + Ky + Rt + Qz. Relict orthopyroxene is locally present (top right corner). Thin section is not oriented. Sample SW13-167 (coordinates: 131.77475, -26.30845; location 14 in Fig. 1 of the main paper).**

**S5 deleted**

**S5 deleted**

---

## Author Response (AR2)

**Author comments (AC) and manuscript changes (MC) on review by editor (Renée Heilbronner) and referee (Florian Fusseis)**

**1)** The rheological evolution of the northern and southern sections of the Woodroffe Thrust is still not very well explained. Simply referring to a follow-up paper is not enough. If the shear zones are strain softening, should not the efficiency of the softening mechanism be the main controlling factor determining the ultimate width? This issue remains unexplained. Why not focus more on the strain softening mechanisms, as suggested? With the key message being that (hydration) reactions control the rheological evolution of a mylonite, the paper would be a clear step forward compared to previous studies by Camacho et al and Bell & Etheridge.

**AC/MC:** We fully understand the point being made here, which is why the planned third paper in the trilogy is devoted to examining the dependence of rheology and microstructure on varying metamorphic conditions and water activity. There is a logical progression to these papers. The first established the overall geometry, kinematics and metamorphic conditions (Wex et al., 2017). The current manuscript (second) documents the preferential distribution of mylonites in the footwall and the variation in water content from south to north. The third considers the variation in microstructure, CPO (EBSD and Texturegoniometer data) and grain size (EBSD) as a basis for establishing the deformation mechanisms and rheology. We could "cut-and-paste" aspects of this third paper but these parts could not be thoroughly developed in the current manuscript, would introduce repetition and overlap with the upcoming paper, and do not really fit the well-defined aim of the current manuscript. We consider that adding limited information on deformation mechanisms without a robust documentation would decrease rather than increase the quality of the current manuscript. We therefore continue to argue against mixing the clearly defined aims of the three planned papers. The reader is referred to a forthcoming paper for a full description of the deformation mechanisms.

**2)** With respect to the  $CO_2$  - your answer to the second reviewers question is not satisfying. If the Woodroffe Thrust did not channelize these fluids, how did the fluids get to where they are now? The hypothesis proposed is not very well supported by the  $CO_2$  data and requires a better discussion.

**AC/MC:** We can only report the observations that we have with regard to the irregular distribution of very minor accessory calcite in otherwise non-carbonaceous rocks. We have now expanded on the discussion to make it clear that the irregular spatial distribution of small amounts and the lack of correlation with indicators for associated introduction of aqueous fluid argue for a local redistribution rather than a pervasive influx. The data from the stable isotope analyses on the calcite are also inconclusive, with the  $\delta^{13}$ C values being partially rock-buffered, reflecting the small amount of fluid involved and the high rock/fluid ratio. We do not insist on publishing this admittedly inconclusive data in the supplementary material but think it is correct to make it available to future workers in the area.

**3)** You did not properly describe (and justify) the thresholding technique you used, error estimates are missing, methods for the separation of peaks are not explained. Statements such as "sophisticated segmentation tools [...] require careful checking of the output results" do not qualify. The reliability of your results needs to be demonstrated - your "believe" that the results are "quite reliable" is not convincing. Looking at the histogram in figure S3, I wonder what motivated the choice of positions of

the boundaries between these phases - as clearly, the peaks are overlapping, and it is virtually impossible to separate Qtz, Plag and Kfsp.

**AC/MC:** We now include the 2 x standard deviation for each of our samples. We also extended the description as to how we estimated the modal abundance of hydrous minerals in our thin sections. The method section in the appendix now provides additional information with regard to our thresholding technique, emphasizing that we did not attempt to separate every single mineral phase from one another, but only the hydrous from the anhydrous ones. Thresholds were manually defined between the individual grey scale values ranges defined by the hydrous and anhydrous each minerals. In order to achieve a proper separation, we adjusted brightness and contrast in a manner that the grey scale range defined by the hydrous minerals covered a significant portion of the entire spectrum. As a result, the histogram data of the individual anhydrous minerals strongly overlap with one another (as mentioned be the reviewer for the peaks of quartz, K-feldspar and plagioclase in Fig. S3). This approach minimalizes the overlap between the peaks defined by the hydrous minerals with those of the anhydrous minerals and thus provides a proper basis for the desired separation. We agree with the reviewer that in this case a proper separation of quartz, K-feldspar and plagioclase would not be possible. However, separating individual anhydrous minerals from one another was never our goal.

**A few minor points**

Line 22 - increase in the abundance of fluids

AC/MC: done.

Line 26 - difference between fault and shear zones (cf. Ramsay's definition of shear zones)?

AC/MC: Reference added.

**Line 53 onward** - I think this could be pitched better - reads a bit like "just another contribution" and doesn't really grab the reader's attention/interest. Why not emphasize what new data you are adding?

**AC/MC:** We added the fact that we utilize the radiogenic signature of footwall and hanging wall rocks to constrain their respective degree of reworking the ductile mylonite zone.

Line 377 - For the sake of whose simplicity?

**AC/MC:** We removed this statement.

**Line 447** - *This is avoiding the important question as to where these fluids come from and how they travelled.*

**AC/MC:** We cannot answer these questions, as the stable isotope data is inconclusive.

Line 518 - and increasing displacement?

**AC/MC:** According to Hull (1988), shear zone thickness decreases with decreasing displacement.

Line 535 - Why is this a contradiction?

**AC/MC:** The sentence has been rewritten for greater clarity and the word "contradiction" no longer appears.

Line 539 - effective viscosity contrast?

AC/MC: Rephrased.

Line 545 - It's not exactly new that even small amounts of fluid will support strain localization.

**AC/MC:** True, but we do not claim that we were the first to come up with this conclusion. We simply use it as an argument why the shearing is preferentially developed in the footwall, because it has a slightly higher water content than the hanging wall in an overall relatively dry environment – and this explains the "inverted" or "unexpected" distribution of mylonites between hanging wall and footwall.

[revised manuscript text omitted]